# Application of the Fractional Riccati Equation for Mathematical Modeling of Dynamic Processes with Saturation and Memory Effect

Dmitriy Tverdyi [1,2,*] and Roman Parovik [1,3]

1   Integrated Laboratory "Natural Disasters of Kamchatka-Earthquakes, Volcanic Eruptions", Vitus Bering Kamchatka State University, Pogranichnaya 4, Kamchatka Region, 683032 Petropavlovsk-Kamchatskiy, Russia; parovik@ikir.ru
2   Laboratory of Electromagnetic Radiation, Institute of Cosmophysical Research and Radio Wave Propagation, Far Eastern Branch of the Russian Academy of Sciences, Mirnaya 7, Kamchatka Region, 684034 Paratunka, Russia
3   Laboratory of Physical Process Modeling, Institute of Cosmophysical Research and Radio Wave Propagation, Far Eastern Branch of the Russian Academy of Sciences, Mirnaya 7, Kamchatka Region, 684034 Paratunka, Russia
*   Correspondence: dimsolid95@gmail.com

**Abstract:** In this study, the model Riccati equation with variable coefficients as functions, as well as a derivative of a fractional variable order (VO) of the Gerasimov-Caputo type, is used to approximate the data for some physical processes with saturation. In particular, the proposed model is applied to the description of solar activity (SA), namely the number of sunspots observed over the past 25 years. It is also used to describe data from Johns Hopkins University on coronavirus infection COVID-19, in particular data on the Russian Federation and the Republic of Uzbekistan. Finally, it is used to study issues related to seismic activity, in particular, the description of data on the volumetric activity of Radon (RVA). The Riccati equation used in the mathematical model was numerically solved by constructing an implicit finite difference scheme (IFDS) and its implementation by the modified Newton method (MNM). The calculated curves obtained in the study are compared with known experimental data. It is shown that if the model parameters are chosen appropriately, the model curves will give results that correlate well with real experimental data. Moreover, with other parameters of the model, it is possible to make some prediction about the possible course of the considered processes.

**Keywords:** Riccati equation; fractional derivative of variable order (VO); Gerasimov-Caputo derivative; finite-difference schemes; mathematical modeling of dynamic processes; saturation and memory effect; COVID-19; solar activity; radon accumulation



## 1. Introduction

The Riccati equation is well-known and of great interest, which is confirmed by a large number of studies, in particular, its version with a noninteger (fractional) derivative. This topic has shown increasing interest of researchers over the past 30 years, both in theoretical issues [1,2] and many other articles, and there are many applications of the Riccati equation [3] in many sciences. For example: in economics, to simulate volatility in the financial market [4]; in physics, to examine questions of inelastic hysteresis of wave processes in media with losses and saturation [5]; and in the science of epidemiology, which is quite relevant today—in particular, it is used by the authors of [6] to build logistic models of the epidemic, to determine the time to reach a plateau and decline in the process.

However, before we move on to the fractional Riccati equation considered in the work, we should give an insight into some issues and concepts that are closely related to our problem.

Vito Volterra is one of the pioneers in questions of the memory effect and possible practical applications of the phenomenon of heredity, in particular to problems arising in physics. In the books [7,8], however, V.Volterra himself notes that in fact the concept of memory was introduced into physics by Pekar in 1907. Moreover, according to the work of Uchaikin V.V. [9], such phenomena as fatigue of metals, delayed waves and some other hereditary processes with some delay in time were known even earlier.

The effects of memory or heredity (in other words, hereditarity) may have various processes with saturation, this may indicate the presence of cause-and-effect relationships in the dynamics of such processes. In other words, the concept of heredity means that the system stores information about its history and can then rely on it. Mathematically, this phenomenon can be described using integro-differential equations, where the difference integrands are memory functions [7,9].

When choosing memory functions—as power functions—we easily and naturally pass to the mathematical apparatus of fractional calculus [10,11], namely, to derivatives of fractional order [12–14], and also derivatives of fractional variable order, or simply "variable order" VO according to [15,16]. Fractional differentiation and fractional integration are generalizations of integer-order differentiation and integration, and include $n$-th derivatives and $n$-fold integrals, where $n \in N$, as special cases.

By and large, fractional calculus is a well studied, with many applications, and is a prominent part of mathematical theory. Apart from being included in various systems of equations in the form of differentiation operators, it is also of considerable interest to many mathematicians. The study of this subject has been going on for more than 300 years and to this day. Fractional calculus has attracted the interest of many scientific minds for at least the last 50 years, from the time of [17,18] to the present day. The study of this issue is associated with the following names: A. Nakhushev [10], V. Uchaikin [9,13], A. Pskhu [19], A. Kilbas [12], O. Mamchuev [20], and R. Parovik [21].

Given the above, it becomes clear that the fractional Riccati equation is called the classical Riccati equation but with a fractional derivative. By analogy with the idea of differentiation and integration mentioned earlier, the fractional Riccati equation is also a generalization of the classical Riccati equation. Furthermore, the introduction of another degree of freedom—the order of the fractional derivative—allows one to obtain a of the experimental data of processes with saturation, more flexible than using the classical equation. Moreover, if a fractional derivative of a variable order is introduced into the Riccati equation, then this will make it possible to describe the experimental data even more flexibly. This, in fact, we will show in this study.

As noted at the beginning, the fractional Riccati equation has been increasingly of interest to researchers in recent years, including us, otherwise this work would not exist. Relatively recently, somewhere in the early 2000s, the first works on the study of the fractional Riccati equation appear. This article is not a review article and therefore we are forced to skip the literature review on this subject. However, in our opinion, some of the works that influenced this study should be briefly mentioned. The work [22] in 2012 by Sweilam, N.H., Khader, M.M., and Mahdy, A.M.S. is of particular importance, since it was this that attracted one of the authors, Tverdyi, D.A., to study the fractional Riccati equation. There is also the work of the authors Min Cai and Changpin Li [23] in 2020, which helped with the theoretical justification of numerical schemes for fractional derivatives of VO.

A more detailed analysis of the literature on the research topic is carried out by the authors in the previous article [24], which is the theoretical justification for this article. The following conclusions can be briefly drawn from the analysis:

- comparison is rarely made between experimental data of processes with saturation and calculated data on a mathematical model;
- the order of the fractional derivative is often constant, which may not be enough for our purposes;
- the issues related to the numerical solution of the fractional Riccati equation with variable fractionality of the derivative are poorly studied.

Additionally, as part of the research, the issues of stability and convergence of the numerical solution are considered in the author's work [24].

It is known that the introduction, at one time, of derivatives and partial derivatives made it possible to describe much more complex phenomena, and in a more accurate way. Similarly, the introduction of the fractional derivative allowed some processes to be described more flexibly, implying the presence of a memory effect in them, which is described by fractional operators. As a consequence, similarly, the introduction of a derivative of a (fractional) variable order makes it possible to describe some physical processes with memory even more flexibly, but now implying the fact that memory may not be constant and depend on time.

In the same article, questions arose related to the use of the developed numerical scheme IFDS for the mathematical modeling of dynamic processes with saturation and memory effect, in order to test the adequacy of the model and in order to find the possible practical application of the proposed fractional model.

A model is considered for describing data on solar activity (SA), expressed in the average monthly number of sunspots, in order to give some forecast for the intensity of this process, and, as a consequence, the approximate boundaries of the current and future SA cycle.

A mathematical modeling of the spread of COVID-19 infection is proposed, which takes into account many possible factors that can affect the change in the number of cases of infection. However, it should be noted that the authors could not have taken into account all the factors. The model describes trends: by new cases of infection, and by the total number of infected, both in the Russian Federation and in the Republic of Uzbekistan [25].

Also with the help of mathematical modeling the process of radon accumulation in the chamber with gas-discharge counters [26] is investigated, i.e., RVA, in order to develop a methodology for predicting strong earthquakes based on continuous monitoring of radon volumetric activity [27] in the subsoil air with a high degree of detail.

In terms of searching for applications of the mathematical model, a "phenomenological" approach to the description of experimental data was chosen. Namely, this approach, a model is created for the observed phenomena, in which they do not pay attention to the actually occurring processes of a "lower" level.

## 2. Some Definitions of Fractional Calculus and Hereditarity, and Their Mutual Connection

Consider the hereditary (memory effect) equation:

$$\int_0^t K(t-\sigma,t)\dot{u}(\sigma)d\sigma - b(t)u(t) = f(u(t),t), \tag{1}$$

where $u(t) \in C^1[0,T]$ is the solution function, $K(t-\sigma,t)$ the sub integral memory function, $t \in [0,T]$ the current time, $T > 0$ the total simulation time, $b(t) > 0$ is a continuous function, and $f(u(t),t)$ is also a continuous function, which satisfies the Lipschitz condition with respect to the variable $u(t)$, as follows: $|f(u_1,t) - f(u_2,t)| < L|u_1 - u_2|$.

**Remark 1.** *V. Voltera [7], in his works defines heredity on the interval $(-\infty, t)$, that is, considers complete heredity. In this paper, we will consider bounded hereditarity defined on the subinterval $(0, t)$.*

It is a well-known fact that the process under consideration has no memory if $K(t-\sigma) = \delta(t-\sigma)$ is a Dirac function, and vice versa, the process has full memory if $K(t-\sigma) = H(t-\sigma)$ is the Heaviside function.

If we use a power function as a memory function, then an intermediate case arises, and we can go to the mathematical apparatus of fractional calculus [10,11], namely, to fractional order derivatives [12–14,17].

**Remark 2.** *Power-laws are widespread in various branches of science and knowledge. A feature of power-law memory is that the process gradually "forgets" about its prehistory.*

**Definition 1.** *Let the memory function be defined as:*

$$K(t - \sigma, t) = \phi^{\alpha(t)-1} \frac{(t - \sigma)^{-\alpha(t)}}{\Gamma(1 - \alpha(t))}, \qquad 0 < \alpha(t) < 1, \tag{2}$$

*$\alpha(t)$ is a function responsible for the intensity of the process under study, $\phi$—some process constant [28].*

**Remark 3.** *Moreover, dependence on t leads us to the memory variable phenomenon. In practice, this allows VO to be more flexible, to describe some physical processes with memory, than the usual non-variable fractional derivative.*

**Definition 2.** *Operator of fractional variable order $0 < \alpha(t) < 1$, acting on a function $u(t) \in C[0, T]$:*

$$\partial_{0t}^{\alpha(t)} u(\sigma) = \frac{1}{\Gamma(1 - \alpha(t))} \int_0^t \frac{\dot{u}(\sigma)}{(t - \sigma)^{\alpha(t)}} d\sigma, \tag{3}$$

*where $\dot{u}(t) = \frac{du}{dt}$, and $t \in [0, T]$—current time, $T > 0$—simulation time, $\Gamma(x) = \int_0^\infty e^{-t} t^{x-1} dt$, $x \in \mathbb{C} : R(x) > 0$—Euler's gamma function, will be called the derivative of a VO $0 < \alpha(t) < 1$ Gerasimov-Caputo type [29–31].*

Currently, there are dozens of definitions of the fractional derivative, but in this manuscript we will understand it in the sense of Definition 1. For more information about the fractional order operator (3), see [14].

**Remark 4.** *A fairly well-known (ordinary) Gerasimov-Caputo operator [29–31], which has a constant fractional order $\alpha$, can be obtained from the Gerasimov-Caputo "type" operator (3), simply by discarding in $\alpha(t)$ depending on the time t.*

**Remark 5.** *Most authors from the CIS countries and Russia refer to the VO operator (3) exactly as the Gerasimov-Caputo operator. The authors of this article follow the same tradition. More about this can be found in [24], adjacent to this work. At the same time, many authors from all over the world call the Formula (3) a fractional derivative in the sense of Caputo.*

Then, taking into account (2), we can write the Equation (1) in terms of the derivative of the Gerasimov-Caputo type (3) as a fractional equation:

$$\phi^{\alpha(t)-1} \partial_{0t}^{\alpha(t)} u(\sigma) - b(t)u(t) = f(u(t), t),$$

Let us set the process constant $\phi = 1$. Looking ahead, we can say that, based on the simulation results for specific applications, this assumption turned out to be correct. Then:

$$\partial_{0t}^{\alpha(t)} u(\sigma) - b(t)u(t) = f(u(t), t), \tag{4}$$

**Definition 3.** *Equations of the form (4) with derivatives of variable fractional order (3) will be called fractional equations.*

## 3. Formulation of the Problem

Consider the fractional Riccati equation [32]. To do this, in the Equation (4) we put $f(u(t), t) = -a(t)u(t)^2 + c(t)$. Then we can proceed to the following Cauchy problem:

$$\partial_{0t}^{\alpha(t)} u(\sigma) + a(t)u^2(t) - b(t)u(t) - c(t) = 0, \qquad u(0) = u_0, \tag{5}$$

where $u(t) \in C^2[0, T]$—solution function, $u_0$—a given constant, $t \in [0, T]$—current time, $T > 0$—simulation time, $a(t), b(t), c(t)$—given continuous functions on the segment $[0, T]$, and $b(t) > a(t)$.

**Definition 4.** *The equation in (5) will be called the fractional Riccati equation.*

### 4. Solution Method

Since the Cauchy problem (5) is nonlinear, then using the methods of [33–35] finite difference schemes, we will find a numerical solution to the problem.

We assume that the discretization grid is uniform, therefore we divide the segment $[0, T]$ into $N$ grid nodes in equal parts with the sampling step $\tau = T/N$. As a result, $u(t) \in C^2[0, T]$, the solution function, on the grid, will take the form of a grid analogue: $u(t_k)$ or $u_k$, where $k = 1, \ldots, N$. Similarly, the function $0 < \alpha(t) < 1$ will go to: $\alpha(t_k)$ or $\alpha_k$. Similarly, this is also true for: $a(t), b(t), c(t)$—given continuous functions of the Riccati equation on the interval $[0, T]$.

Replacing the variable order fraction in Equation (5) by its approximation, according to [24], we obtain a discrete analogue of the Cauchy problem for the fractional Riccati equation, in the form of an IFDS:

$$A_k \sum_{j=0}^{k-1} w_j^k \left( u_{k-j} - u_{k-j-1} \right) + a_k u_k^2 - b_k u_k - c_k = 0,$$

$$A_k = \frac{\tau^{-\alpha_k}}{\Gamma(2 - \alpha_k)}, \qquad w_j^k = (j+1)^{1-\alpha_k} - j^{1-\alpha_k}, \tag{6}$$

$$k = 1, \ldots, N, \qquad u_0 = C,$$

where $C$ is a known constant.

As a result, we obtain a system of nonlinear algebraic equations represented by the scheme (6). To solve it, we will use the MNM [24,33,36] method. It is known that the ordinary Newton method (ONM) requires the calculation of the Jacobian matrix (that is, the solution of a system of nonlinear algebraic equations) at each iteration. Using MNM allows us to calculate the Jacobian only once in the initial approximation $x_0$, but this leads to a decrease in the order of convergence of the method (compared to ONM) to the first order.

The issues of stability and convergence of the numerical solution are considered in the author's work [24]. Where the following is shown:

- For the numerical solution of the Cauchy problem (5), nonlocal numerical schemes are proposed: EFDS and IFDS. To solve IFDS, the MNM method is considered with a modification according to (6);
- Questions about the stability, convergence of methods and approximations of a fractional operator are considered. The corresponding theorems are proved;
- It is established that: IFDS is unconditionally stable and has order of convergence $O(\tau^{2-\widehat{\alpha}})$, where $\widehat{\alpha} = \max_k(\alpha(t_k))$, EFDS—conditionally (for $\tau \leq (2^{1-\widehat{\alpha}} - 1)/\bar{b}, \bar{b} = \max_k(b(t_k))$) is stable and has order convergence $O(\tau)$;
- Test examples and estimates of the accuracy of calculations according to the Runge rule confirm the theoretical conclusions;
- From the set of test cases, it appears that the IFDS scheme has a maximum error that is an order of magnitude smaller than that of EFDS. Moreover, IFDS is somewhat more accurate than EFDS.

The above, and the last point in particular, will govern the choice of an IFDS scheme.

**Remark 6.** *Since the EFDS circuit in question (6) conventionally converges with the first order [24]. That, as an initial approximation for MNM, to solve an IFDS, you can take the last value of $u_k$, obtained according to the EFDS scheme when the condition of the EFDS convergence is performed.*

Additionally, according to the results of test cases [24], for practical use, it is important to note:

- If the coefficients in the model equation vary by harmonic law, their form resembles the form of curves for vibrational processes;
- It can be seen from the figures that the calculated curves for the proposed model can have an s-shaped shape, which is typical for dynamic processes in saturated media;
- It is also seen that the trend of the calculated curves increases as the steady state is reached;
- This dynamic occurs in the economy when describing cycles and crises [37].

## 5. Software Package for MAPLE

In this section, we present a software package for the MAPLE 2021 symbolic mathematics environment, designed to solve problems: mathematical modeling, comparison of the result with real experimental data, and visualization of the results. This software package includes: an executable file that implements the processes described above and defines the program logic, and the previously developed «FDREext» library containing the functions necessary for research.

Next, as an example, we will present the code of the executable file step-by-step , using Example 3 for the Simulation the dynamics of infection with coronavirus COVID-19 task, discussed in the Section 8.

For the COVID-19 simulation task Section 8, all stages (subprograms) of the generalized block diagram shown in the Figure 1 are executed. However, for RVA and SA simulation tasks, only a subprograms is needed.

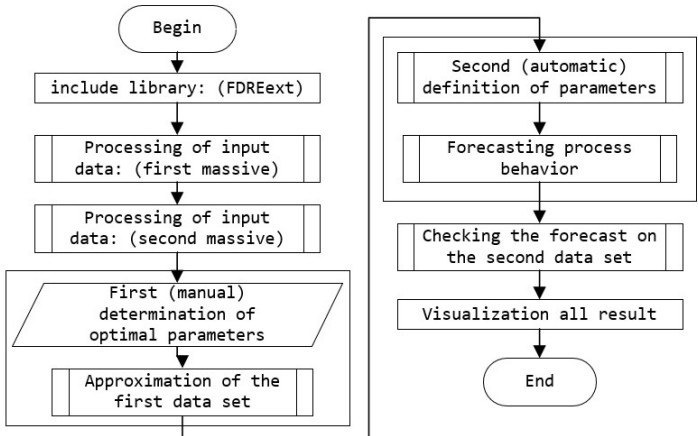

**Figure 1.** Generalized block diagram of the software package.

### 5.1. Pluggable Subprograms

This library (subroutine) contains: procedures for the numerical analysis of the fractional Riccati equation with a variable fractional order of the Gerasimov-Caputo type derivative and nonconstant coefficients, and with the ability to visualize the results, more details in [24]. Program code for «FDREext» has [38] certificate of state registration of computer programs. The library for the solution is called using the instructions shown in the Figure 2:

```
>  restart,
>  with(FDREext);
[ApproxFractDeriv, ClearSeq, CreateFileName, CreateFilePath, ExcelExtractColumn,
    ExtendedData, NormalizeOnMax, NormalizeOnValue, PeriodicFunc, PlotData,
    ResearchOnError, TextExtractColumn, WriteFile_txt, WriteFile_xlxs]
```

**Figure 2.** Library call «FDREext».



**Remark 7.** *Further, the variables, the result of the function and some of the arguments of the functions, will be specified without a specific data type. This will not cause errors in most cases, since when processing input arguments, they are converted to the desired type (for the convenience of the user), if it is allowed. Some arguments of the functions implemented in the library, it is better to specify already in the required data type, and we will indicate this when describing some functions. However, note that the purpose of this section is not to describe in detail the functionality of the «FDREext» library.*

*5.2. Processing of Input Data*

Some «FDREext» functions, for convenience, work with global variables, such as the variable: `Path` , which will contain the path to the experimental data file, including its name and extension:

```
Path := cat(CreateFilePath(), "Covid_Select_Data_UZB_(to_16.09.2021).xlsx")
```

The library implements the ability to work with input/output files with extensions: `.txt, .xlsx`.

Next, you need to read the file and extract the necessary data into the `input_data` variable. The following code will extract from the `.xlsx` file the column for the given keyword, which is contained in the first line of the file. It will also create a few auxiliary variables:

```
key_column := "new_cases";
input_data := ExcelExtractColumn(key_column, set_emptycell = undefined):
num_elems := nops([input_data]);
input_data_new_cases_1 := input_data:
max_1_input_data_new_cases := max([input_data_new_cases_1]);
```

**Remark 8.** *It is important to remember that the MAPLE system considers the dataset:* $1, 2, 3, 4, \ldots$ *to be the data type* `exprseq`*, i.e., "sequence". However, the data set* $[1, 2, 3, 4, \ldots]$ *is a* `list` *data type, and many MAPLE functions work with the* `list` *type. We, in the development of «FDREext», adhered to the same principle.*

Now, let us do the same for the second data file:

```
Path := cat(CreateFilePath(), "Covid_Select_Data_UZB_(to_31.12.2021).xlsx");
input_data := ExcelExtractColumn(key_column, set_emptycell = undefined):
num_elems := nops([input_data]);
input_data_new_cases_2 := input_data:
max_2_input_data_new_cases := max([input_data_new_cases_2]);
```

In the future, in order to compare the experimental data with the model data, in most cases we will need to normalize the array of experimental data to the maximum. Consider an example, let `input_data_new_cases_1` be an array of data up to 16 September 2021, and `input_data_new_cases_2` be an array of data up to 31 December 2021. `input_data_new_cases_2` has a maximum element with a value greater than the maximum element from `input_data_new_cases_1`. Then you need to normalize both arrays to the maximum element with a large value.

In order not to have the situation as in (Figure 3), run the following code:

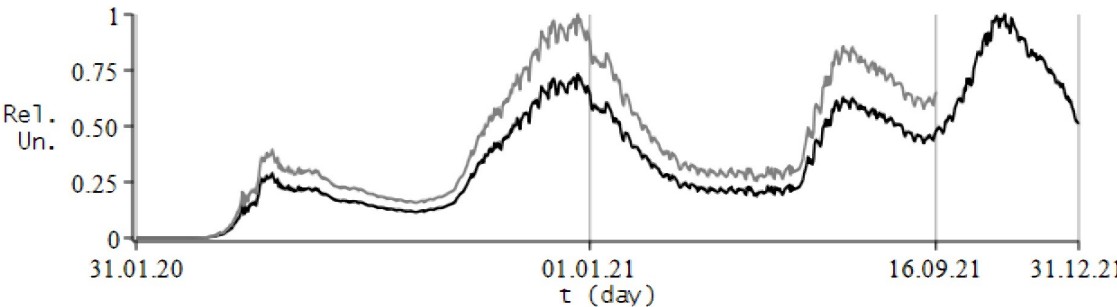

**Figure 3.** An example of a case of incorrect normalization of two data sets.

```
if (max_2_input_data_new_cases <= max_1_input_data_new_cases) then
    value_for_normalize_data := max_1_input_data_new_cases;
else
    value_for_normalize_data := max_2_input_data_new_cases;
end if;
```

as a result, we get the normalization factor `value_for_normalize_data` for the first data array.

Now, you can display the input data on the chart:

```
set_labels := ["t (day)", key_column]; set_color := grey;
set_linestyle := solid; set_type_plot := line;
set_legend := typeset("[data]. Num point = %1. (Input data from (.xlxs)
                file).
                %2 (to_16.09.2021)", num_elems, key_column):
plot_new_cases_1 := PlotData([input_data_new_cases_1], nops([input_data_
                new_
                cases_1]), nops([input_data_new_cases_1]),
                type_plot = set_type_plot, type_linestyle =
                set_linestyle, type_color = set_color,
                type_labels = set_labels, type_legend =
                set_legend)
```

Let us normalize for the first data set:

```
data_norm_1 := NormalizeOnValue([input_data_new_cases_1], value_for_
normalize_data):
max_data_norm_1 := max([data_norm_1]);
num_elems := nops([data_norm_1]);
data_1 := data_norm_1;
```

and, if desired, you can display the result on a chart using the `PlotData` function, which has already been used previously.

Additionally, in «FDREext», there is a function for smoothing/reducing data. Smoothing may be needed if, for example, some data fragments are missing, or we would like to have a denser data stream, that is, for example, 100 parameter measurements every 1 h, smooth up to 150 measurements every 45 min. Reduction, on the contrary, selects only a part of them from the data set with some equal frequency, while maintaining the trend of course. If necessary, then you need to uncomment the first line of the following code:

```
#need_extend := "No";
```

```
if (need_extend <> "No") then
    num_elems_before_1 := nops([data_1]);
    set_dimension_interpolation_1 := round(num_elems_before_1/2);
```

```
    data_extended_1 := ExtendedData([data_1], set_dimension_interpolation_1,
                             type_interpolation = cubic);
num_elems_1 := nops([data_extended_1]);
    data_1 := data_extended_1;
    set_labels := ["t (day)", "Rel.\n Un."]; set_color := green;
set_linestyle := solid; set_type_plot := line;
set_legend := typeset("(Extended)_[data_1]. Num point = %1. (Normalize
                       on max).
                       %2 (to_16.09.2021)", num_elems_1, key_column);
    plot_extrapolaton_1 := PlotData([data_extended_1], num_elems_1, num_
                          elems_
                          before_1, type_plot = set_type_plot,
                          type_
                          linestyle = set_linestyle, type_color =
                          set_color, type_labels = set_labels,
                          type_legend = set_legend);
    plot_data_1 := plot_extrapolaton_1;
    plots[display](plot_norm_1, plot_extrapolaton_1); print(%);
end if:
```

Let us normalize for the second data set:

```
data_norm_2 := NormalizeOnValue([input_data_new_cases_2], value_for_
normalize_data):
max_data_norm_2 := max([data_norm_2]);
num_elems := nops([data_norm_2]);
data_2 := data_norm_2;
```

Similar to the case with experimental data, for comparison, in most cases we will need to normalize to the maximum, as well as the array containing the simulation results. However, after normalization, this array will need to be multiplied by a special coefficient `value_for_normalize_result`, so as not to have a situation where the experimental data along the vertical axis are distributed on the segment $[0, 0.7]$, and the data of the simulation result on the segment $[0, 1]$. To do this, we will execute:

```
if (max_data_norm_1 <= max_data_norm_2) then
    value_for_normalize_result := max_data_norm_1;
else
    value_for_normalize_result := max_data_norm_2;
end if;
```

as a result, we obtain the normalization factor `value_for_normalize_data` for the second data array.

### 5.3. Approximation of the First Data Set

For the proposed numerical scheme (6), model (5), you need to set the following parameters:

$$N = 551, \quad T = 551, \quad u_0 = 0.0001,$$

$$\alpha_k = 0.98 \sin\left(\frac{1.64\pi kh}{T}\right)^2, \quad a_k = \frac{kh}{T}, \quad b = 0.25, \quad c_k = 0.5 \sin\left(\frac{\exp(7kh/T)kh}{T^2}\right)^2 \quad (7)$$

main parameters $N, T$—here are determined by the dimension of the first data array.

**Remark 9.** *As already mentioned in the introduction, the "phenomenological" approach to modeling is chosen, and therefore the model parameters $\alpha(t)$ and $a(t), b(t), c(t)$ for approximating the first data set will be determined "manually". This is because, as a consequence of this approach,*

*we do not know anything about the processes of a "lower" level that would determine the form of these parameters.*

In the code, the main parameters will take the form of local variables:

```
set_T := nops([data_norm_1]);
set_N := nops([data_1]);
set_h := evalf(set_T/set_N);
epsilon := evalf[1](10^(-4));
start_point := data_1[1];
```

and grid functions for the coefficients of the model equation, the form of global variables:

```
a_Func := k*h/T;
b_Func := 0.25;
c_Func := PeriodicFunc(set_function = [sin, 2], set_delta = 0.25,
                       set_theta = 0.5, set_mu = exp(7*k*h/T)/T^2);
Alpha_Func := PeriodicFunc(set_function = [sin, 2], set_delta = 0.49,
                           set_theta = 0.98, set_mu = 1.64*Pi/T)
```

the type of grid functions, to a greater extent from `Alpha_Func`, will also depend on the type of model curves obtained as a result.

The function `PeriodicFunc`—at the output gives some periodic function with given oscillation parameters: `set_delta`—axis, `set_theta`—amplitude, `set_mu`—frequency.

**Remark 10.** *Note that the variable k is the discretization step of the (6) scheme, and at this stage it is only declared, and it can be passed further to functions without a specific value, and already inside functions k will take a specific value in cycles . This will not throw an error since MAPLE is a symbolic calculation system.*

Now, let us execute the function for the numerical solution of the model equation:

```
approx_result_1 := ApproxFractDeriv(numerical_method = "IFDS",
                                    start_iter_MNM = "Last_EFDS",
                                    type_operator = "alpha(t)", set_N,
                                    set_T,
                                    start_point, acccuracy = 10^(-4),info_
                                    print = yes,
                                    graphics_print = yes):
```

where the function arguments, define:

- `numerical_method :: string`—(optional) numeric scheme, default: `"IFDS"`. According to the Section 4, the EFDS and IFDS schemes are implemented;
- `start_iter_MNM :: string`—(optional) method for calculating start iteration for IFDS scheme, see Remark 6, default: `"Last_EFDS"`;
- `type operator :: string`—(optional) modification type for fractional operator (3), default: `"alpha(t)"`;
- `set_N :: integer`—number of nodes of the calculated uniform grid;
- `set_T :: integer`—simulation time;
- `start_point :: anything`—start point, initial value of the Cauchy problem;
- `acccuracy :: float`—(optional) required accuracy of numerical method, default: `"10^(-4)"`;
- `info_print :: string`—(optional) flag for outputting intermediate information about the calculation progress, default: `"No"`;
- `graphics_print :: string`—(optional) flag for outputting intermediate graphic information, default: `"No"`.

arguments (optional), the user can ignore and not set at all, if necessary.

We visualize the obtained numerical solution, compare it on the graph together with the experimental data and calculate the correlation coefficient between them, the result is shown in the Figure 4:

```
approx_result_1 := seq(approx_result_1[i], i = 1 .. nops([approx_result_1])
- 1):
max_approx_result_1 := max([approx_result_1]);
num_elems_approx_result_1 := nops([approx_result_1]);
approx_result_norm_1_0 := NormalizeOnMax([approx_result_1]);
approx_result_norm_1 := NormalizeOnValue([approx_result_norm_1_0],
                                    1/value_for_normalize_result);

coorelation_approx_result_1_with_data := Float(round(Float(evalf(
                                    Statistics[Correlation]
                                    ([data_1],
                                    [approx_result_norm_1])),4)),-4);
set_labels := ["t (day)", "Rel.\n Un."]; set_color := blue;
set_linestyle := solid; set_type_plot := line;
unassign('T', 'h', 'k');
set_legend_1 := typeset("[model curve]. (Correlaton = %2). N = %3, T = %4,
                    [alpha = %5], a = %6, b = %7, c = %8", set_method,
                    coorelation_approx_result_1_with_data,
                    num_elems_approx_result_1, set_T, Alpha_Func,
                    a_Func, b_Func, c_Func);
set_tickmarks_1 := [[1 - 1 = "15.03.20", 294 - 1 = "01.01.21",
                    551 - 1 = "16.09.21"], default];
plot_approx_1 := PlotData([approx_result_norm_1], num_elems_approx_result_1,
                    set_T, type_plot = set_type_plot,
                    type_tickmarks = set_tickmarks_1, type_linestyle =
                    set_linestyle, type_color = set_color,
                    type_labels =
                    set_labels, type_legend = set_legend_1);
plots[display](plot_approx_1, plot_data_1);
```

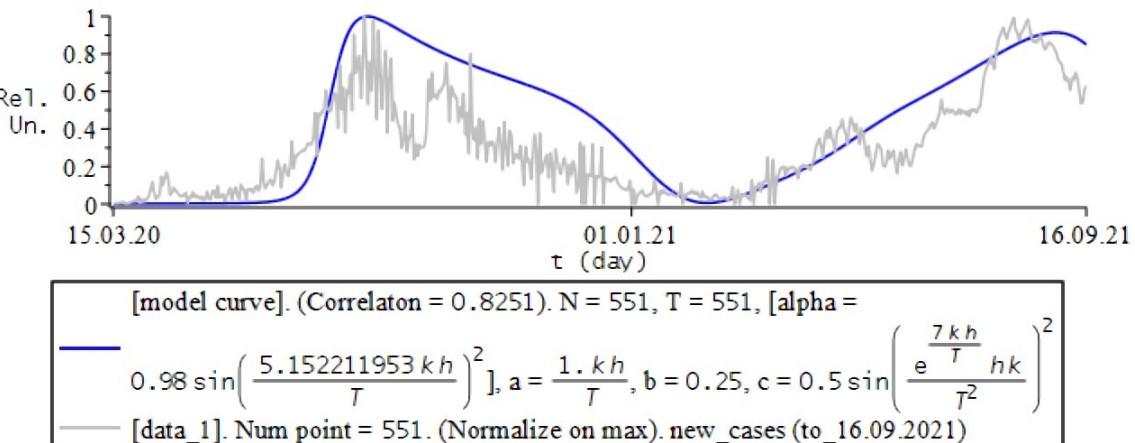

**Figure 4.** The result of executing the code to match.

Let ua denormalize the input data and the result in order to obtain the graphs presented on the initial scale of the experimental data:

```
set_labels := ["t (day)", "New cases of infected"];
```

```
approx_result_norm_1_DE := seq(approx_result_norm_1[i]*value_for_normalize_
                            data,
                            i = 1 .. nops([approx_result_norm_1]));
num_elems_approx_result_1_DE := nops([approx_result_norm_1_DE]);
plot_approx_1_DE := PlotData([approx_result_norm_1_DE], num_elems_approx_
                            result_1_DE,
                            set_T, type_plot = set_type_plot, type_
                            tickmarks =
                            set_tickmarks_1, type_linestyle = set_
                            linestyle,
                            type_color = set_color, type_labels = set_
                            labels,
                            type_legend = set_legend_1);
```

*5.4. Forecasting Process Behavior*

Now, based on the mathematical model (5), with refined modeling parameters based on the approximation of the first data array, we can predict the possible course of the process, in our case, the change in the number of new cases of COVID-19 infection.

Redefine (7) modeling parameters:

$$N = 657, \quad T = 657, \quad u(0) = 0.0001,$$

$$\alpha_k = 0.98 \sin\left(\frac{1.96\pi kh}{T}\right)^2, \quad a_k = \frac{1.1923kh}{T}, \quad b = 0.25, \quad c_k = 0.5 \sin\left(\frac{\exp(8.78kh/T)kh}{T^2}\right)^2 \tag{8}$$

which in code will look like:

```
unassign('add_model_time', 'k', 'set_h', 'set_T', 'set_N');
add_model_time := 106;
add_model_time := 106
set_T := nops([data_norm_1]) + add_model_time;
set_N := nops([data_1]) + add_model_time;
set_h := evalf(set_T/set_N);
unassign('k', 'h', 'T', 'N', 'a_Func', 'b_Func', 'c_Func', 'Alpha_Func'):
a_Func := 1.1923*k*h/T;
b_Func := 0.25;
c_Func := PeriodicFunc(set_function = [sin, 2], set_delta = 0.25, set_theta
                        = 0.5,
                        set_mu = exp(8.78*k*h/T)/T^2);
Alpha_Func := PeriodicFunc(set_function = [sin, 2], set_delta = 0.49,
                            set_theta = 0.98, set_mu = 1.96*Pi/T);
```

**Remark 11.** *It is necessary to redefine the simulation parameters so that the model curve predicting the behavior of the process coincides with the model curve approximating the first data array, in a common area for them.*

Let us solve the model equation again numerically:

```
approx_result_2 := ApproxFractDeriv(numerical_method = "IFDS",
                            start_iter_MNM = "Last_EFDS",
                            type_operator = "alpha(t)", set_N,
                            set_T,
                            start_point, acccuracy = 10^(-4),info_
                            print = yes,
                            graphics_print = yes):
```

Visualize:

```
approx_result_2 := seq(approx_result_2[i], i = 1 .. nops([approx_result_2]) - 1);
max_approx_result_2 := max([approx_result_2]);
num_elems_approx_result_2 := nops([approx_result_2]);
approx_result_norm_2_0 := NormalizeOnMax([approx_result_2]);
approx_result_norm_2 := NormalizeOnValue([approx_result_norm_2_0],
                                1/value_for_normalize_result);
set_labels := ["t (day)", "Rel.\n Un."]; set_color := red;
set_linestyle := solid; set_type_plot := line;
unassign('T', 'h', 'k');
set_legend_2 := typeset("[Prediction model curve]. N = %1+%7, T = %2+%7,
                [alpha = %3],
                  a = %4, b = %5, c = %6", num_elems_approx_result_2 -
                  add_model_time, set_T - add_model_time, Alpha_Func,
                  a_Func, b_Func, c_Func, add_model_time);
set_tickmarks_2 := [[1 - 1 = "15.03.20", 294 - 1 = "01.01.21", 551 - 1 =
                  "16.09.21",
                  (551 - 2) + add_model_time = "31.12.21"], default];
plot_approx_2 := PlotData([approx_result_norm_2], num_elems_approx_result_2,
                  set_T, type_plot = set_type_plot, type_tickmarks
                  =set_tickmarks_2, type_linestyle = set_linestyle,
                  type_color = set_color, type_labels = set_labels,
                  type_legend = set_legend_2);
plots[display](plot_approx_2, plot_data_1);
```

If desired, in a similar way, you can denormalize the input data and the result.

*5.5. Checking the Forecast on the Second Data Set*

Now, let us check how close to reality the predicted (Figure 5, red curve) results turned out to be. To do this, we compare the experimental data from the second array with the simulation results for the parameters (8) specified for the prediction of the first data array.

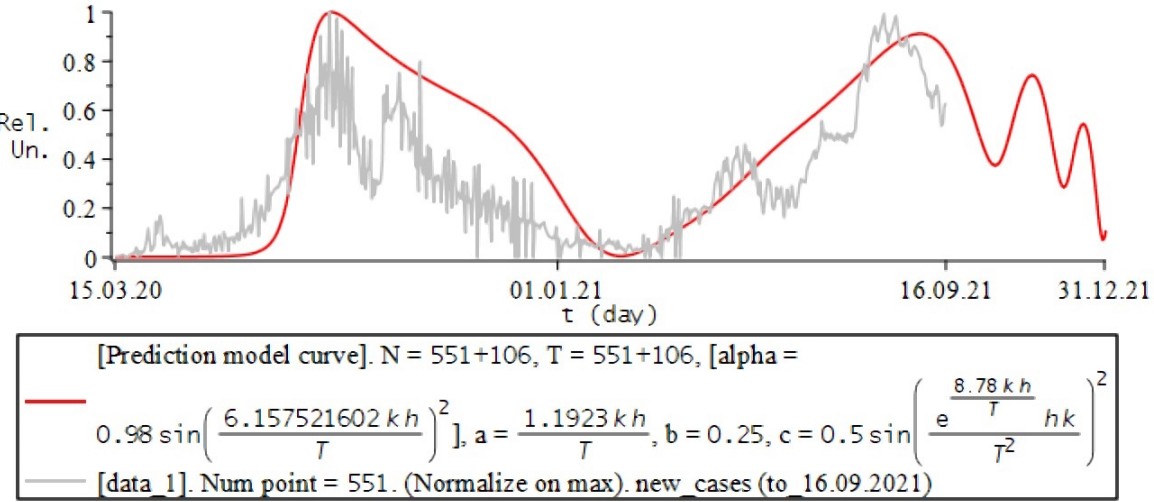

**Figure 5.** Visualization result.

There is no point in calculating a model for approximating the second data array, since now, the dimension of the simulation result for the forecast of the first data array coincides with the dimension of the second data array (after all, we predicted it). For the convenience of continuing the logic of the previous code blocks, let us copy the result of the forecast into a new variable:

```
approx_result_3 := approx_result_2:
```

and now we can check the forecast, for which we calculate the correlation coefficient between them. We visualize this in the same way (as for the forecast), except that the new variables will have the number 3 at the end, for example `plot_approx_3`.

To present the results, it can be convenient to display graphs: approximation, prediction and verification, on one canvas:

```
plots[display](plot_data_2, plot_data_1, plot_approx_3, plot_approx_2,
plot_approx_1);
```

*5.6. «RVAMM» Library, and RVA Simulation*

Additionally, within the framework of the software package for numerical modeling using the fractional Riccati equation, based on «FDREext», the «RVAMM» library (abbreviation for Radon Volumetric Activity Mathematical Modeling), the following was developed to solve the problem of modeling the dynamics of radon accumulation in the accumulative camera discussed in Section 9. The library is called using the instructions shown in the Figure 6:

**Figure 6.** Library call «RVAMM».

The «RVAMM» library is based on the function: `MakarovModelRVA()`, which implements the solution of the classical model (15). The function has the following arguments:

- `A_0_local :: anything`—parameter $A_0$;
- `A_max_local :: anything`—parameter $A_{max}$, due to normalization always $= 1$;
- `lambda_0 :: anything`—parameter $\lambda_0$;
- `set_N :: integer`—number of nodes of the calculated uniform grid;
- `set_T :: integer`—simulation time;

about the meaning of the described parameters from the (15) model, see Section 9 for more details.

By means of «RVAMM», the possibility of automatic selection of modeling parameters is implemented: $\lambda_0$ and $\alpha(t)$ (constant), which is reflected in the following library functions:

1. `LambdaAnalysisRVA()`—the range and step of changing the $\lambda_0$ parameter are specified in the arguments, and using `MakarovModelRVA()` it calculates a number of solutions, from which the best one is selected by the best correlation coefficient with the given experimental data value $\lambda_0$;
2. `AlphaConstAnalysisRVA()`—sets the range and step of changing the $\alpha(t)$ parameter, and using the `ApproxFractDeriv()` function of the «FDREext» library, calculates a number of solutions, from which the best one is selected , according to the correlation coefficient, the value of $\alpha(t)$.

These two procedures are performed on experimental data reduced using the `ExtendedData()` function to increase performance. Moreover, the dimension of the data and the parameter `set_N` must match.

The code of the executable file for the implementation of numerical experiments Section 9.2 of RVA simulation is generally identical to the code described above in Sections 5.2 and 5.3, is schematically represented by the block diagram in the Figure 7, but there are differences:

- only one data file is processed for matching;
- only the approximation stage is used for this task, Section 5.3;
- to automatically override the $\lambda_0$ parameter, `LambdaAnalysisRVA()` is used, and for the fractional model (5) parameter $\alpha(t) = \alpha$, `AlphaConstAnalysisRVA()` will be used , after which the approximation stage Section 5.3 starts again until the optimal parameters are found;
- at the stage of plotting and calculating the correlation, results are needed: both for the proposed fractional model (5) and for the known model (15) implemented in the `MakarovModelRVA()` function.

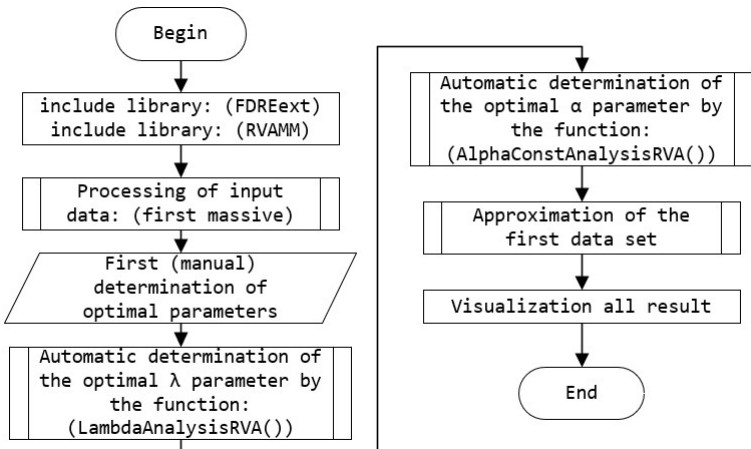

**Figure 7.** Block diagram, executable file for RVA simulation task.

### 5.7. Notes on the Code Used to Simulation SA

The code of the executable file for the implementation of numerical experiments Section 7.2 of SA simulation is generally identical to the code described above in Sections 5.2 and 5.3, is schematically represented by the block diagram in the Figure 8, but there are differences:

- only one data file is processed for matching;
- only the approximation stage is used for this task Section 5.3;
- the parameters $\alpha(t), a(t), b(t), c(t)$ are redefined manually, after which the approximation stage Section 5.3 is started again until the optimal parameters are found.

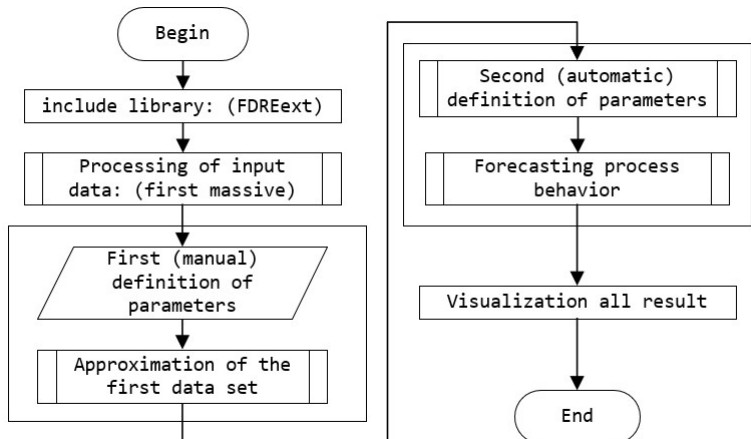

**Figure 8.** Block diagram, executable file for SA simulation task.

The developed program using the functions of the «FDREext» library generalizes the functionality of the previously used other program of the author «MMDCSA» for

modeling the dynamics of solar activity at the ascent stage, for which there are several implementation acts and there is a certificate [39] on state registration computer programs.

## 6. Background to Modeling Processes with Saturation and Memory by the Fractional Riccati Equation

As noted earlier in the introduction, there are few works in which saturation processes are modeled and the calculated data are compared with experimental ones. In this chapter, this thesis can be developed as follows:

- these works mainly have a bias in the physical (real) modeling of the process, but with an attempt to preliminarily build a certain theoretical basis for verification;
- there are even fewer (or almost zero) similar studies for saturation processes and the memory effect;
- there are even fewer works, with a more mathematical bias, involving the apparatus of fractional differentiation and integration, numerical analysis, which would be more in line with the subject and direction of this study.

For example, in the work of the author Landis, C.M. [40], the question of the conditions of deformation saturation of polycrystalline ferroelastic materials is considered. A phenomenological governing law of deformation has been developed; to verify the law, micromechanical self-consistent modeling is carried out on several single crystals. Which shows the adequacy of the proposed model (law) and the fact that such a phenomenological theory is able to capture the complex defining behavior of ferroelastic materials. The work of the authors Pudeleva, O.A. and Semenov, A.S. [41] continues these ideas, but for polycrystalline ferro-piezoceramics, which is a promising multinational material [42] used in sensors, vibration dampers, micromotors, memory elements, and so on.

In [43], authors John M. Bayldon and Isaac M. Daniel conduct a simulation of the flow used in vacuum-based liquid composite molding processes. The authors note that modeling methods for molding liquid composites depends on a good understanding of closely related internal phenomena, and also note a number of important pressure-related and other physical features of this process. Additionally, the fact that the previously presented models, as a result, had a number of assumptions due to which the model was not confirmed in real experiments. The authors also propose an improved compaction model that includes the dependence of compaction pressure on saturation. It is shown that the model gives more physically realistic results.

However, studies close to our work were nevertheless carried out, for example, in the work of Buraev A.V. [44], where the dynamics of solar activity in the period 1998–2010 was studied and its connection with mudflows in the Kabardino-Balkarian Republic was established. Moreover in [44], based on data on SA dynamics, it is shown that the rise and fall of SA occur along a curve very close to the generalized logistic curve [45–47], and this process is nonlinear and fractal [48,49] . However, in this work, the model Riccati equation with a fractional derivative was with constant coefficients.

In our study, based on the physical assumptions of the nonlinearity and fractality of the process, mathematical models (5) are built based on the fractional Riccati equation, since the Riccati equation well describes processes that obey the logistic law [50], and an arbitrary order of the fractional derivative gives a wide range for refining the mathematical model with saturation and takes into account the effect of the variable memory of the dynamic system.

## 7. Simulation the Dynamics of Solar Activity

### 7.1. Formulation of the Problem

Studies of solar-terrestrial connections—the reactions of the outer shells of the Earth, including the biosphere—to changes in solar activity, have been actively pursued throughout the 20th century. The Earth receives not only light and heat from the Sun, but is also exposed to UV and X-ray radiation, solar wind, etc. Variations in the power of these factors with a

change in the level of solar activity cause a chain of phenomena in interplanetary space, the outer shells of the Earth, and changes in the magnetosphere are especially noticeable.

A link is statistically established: an increase in the number of accidents in the United States of America power grids close to the auroral zone in the wake of the level of geomagnetic activity. However, in years of minimal activity, the probabilities of accidents in hazardous and safe areas are practically equalized [51]. It is also assumed that there is a connection between the level of solar and geomagnetic disturbance and the course of a number of processes in the Earth's biosphere, such as: the dynamics of animal populations, epidemics, the number of cardiovascular crises, etc.

The study of solar-terrestrial relations is not only a fundamental scientific problem, but also has great predictive value. Forecasts of the state of the magnetosphere and other shells of the Earth are extremely necessary for solving practical problems in the field of astronautics, radio communications, transport, meteorology and climatology, agriculture, biology and medicine.

The strongest and most noticeable manifestation of solar activity (SA) is a powerful flare on the Sun, and the consequences of the flare begin to affect the near-Earth space relatively quickly. In particular, particles accelerated in a flare, invading the ionosphere and stratosphere of polar latitudes, cause a long-term deterioration of short-wave radio communications, for 10s of hours, and contribute to the depletion of the ozone layer. The solar flare itself is caused by the compression of plasma under the influence of a magnetic field, hence the interest in observing the activity of the Sun's magnetosphere.

Spots on the Sun are an obvious sign of its activity, the formation of which is associated with the Sun's magnetic field. After 17 years of observations, Heinrich Schwabe found that the number of sunspots changes over time. Years of minimum sunspots on the surface of the Sun may be zero; in years of maximum sunspots, their number is measured in tens. Highs and lows alternate on average every 11 years (from 7 to 17 years); however, there may be longer SA cycles.

Since the change in the number of sunspots is the most studied type of solar activity, we can apply the described (5) model to approximate data on the average monthly number of sunspots in the period from May 1996 to August 2021, i.e., 23, 24 and beginning of 25–SA cycles, with a step of 1 month.

The data on the CA process presented in the graph in Figure 9, were obtained from the World Data Center database for the production, preservation and dissemination of the international sunspot number, from the Royal Belgian Observatory website, project: Sunspot Index and Long-term Solar Observations [52], and published on the website www.sidc.be (accessed on 15 August 2021).

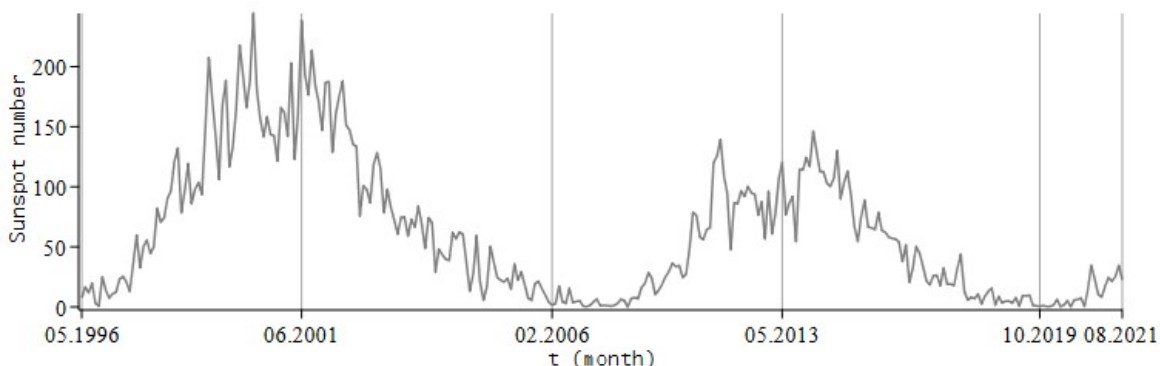

**Figure 9.** Monthly sunspot data from May 1996 to August 2021 in 1-month increments.

### 7.2. Numerical Experiment for SA

The modeling parameters used were calculated with different variations in the values of the modeling parameters, from which the most appropriate combination was selected according to the maximum correlation coefficient with the experimental ones.

For the input data, we will normalize the number of sunspots to the maximum; therefore, the experimental data are in relative units. The simulation results were not standardized.

**Example 1.** *We will approximate (Figure 10, blue curve) the input data with the model (5), with the parameters:*

$$N = 304, \quad T = 304, \quad u_0 = 0.031109,$$

$$\alpha(t) = -0.5 \exp\left(\frac{-t}{T}\right) \cos\left(\frac{2.2\pi t}{T}\right)^2 + 0.25 - \frac{0.125t}{T} + 0.25 \exp\left(\frac{-t}{T}\right),$$

$$a(t) = \frac{2.85t}{T}, \quad b = 0.01,$$
$$(9)$$

$$c(t) = -0.5 \exp\left(\frac{-t}{T}\right) \cos\left(\frac{2.2\pi t}{T}\right)^2 + 0.25 - \frac{0.125t}{T} + 0.25 \exp\left(\frac{-t}{T}\right).$$

*Let us carry out the proposed model (5), forecasting (Figure 10, red curve) the average monthly sunspot number for 10 years, from August 2021 to August 2031, with the parameters:*

$$N = 424, \quad T = 424, \quad u_0 = 0.031109,$$

$$\alpha(t) = -0.5 \exp\left(\frac{-1.3947t}{T}\right) \cos\left(\frac{3.068\pi t}{T}\right)^2 + 0.25 - \frac{0.1743t}{T} + 0.25 \exp\left(\frac{-1.3947t}{T}\right),$$

$$a(t) = 1.394\frac{2.85t}{T}, \quad b = 0.01,$$
$$(10)$$

$$c(t) = -0.5 \exp\left(\frac{-1.3947t}{T}\right) \cos\left(\frac{3.068\pi t}{T}\right)^2 + 0.25 - \frac{0.1743t}{T} + 0.25 \exp\left(\frac{-1.3947t}{T}\right).$$

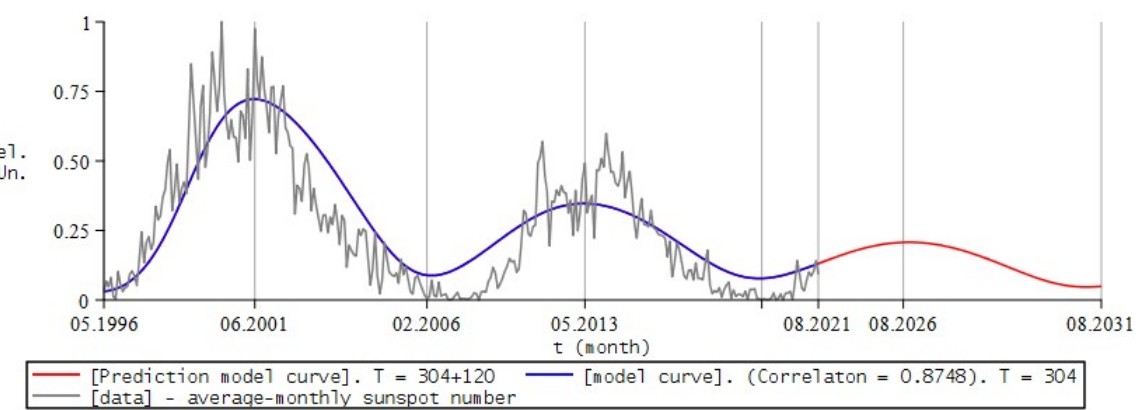

**Figure 10.** Numerical experiment for SA. Correlation coefficient 87.4%.

*7.3. Comparison and Conclusions*

It is shown that with the optimal (9) choice of the corresponding simulation parameters: $\alpha(t)$ and $a(t), b(t), c(t)$, the calculated curves are in good agreement with the smoothed experimental data for SA cycles. Additionally, with slightly different parameters (10), the model (5) is able to give some forecast of the possible number of sunspots and, as a result, approximate boundaries of the current and future SA cycle. It has been shown that solar activity is decreasing.

Furthermore, from (Figure 11) it can be seen that the results obtained by the mathematical model (5) are in good agreement with the known models and the results of solar activity forecasts. The graphs in (Figure 11) are provided by the site www.spaceweatherlive.com, accessed on 10 February 2022.

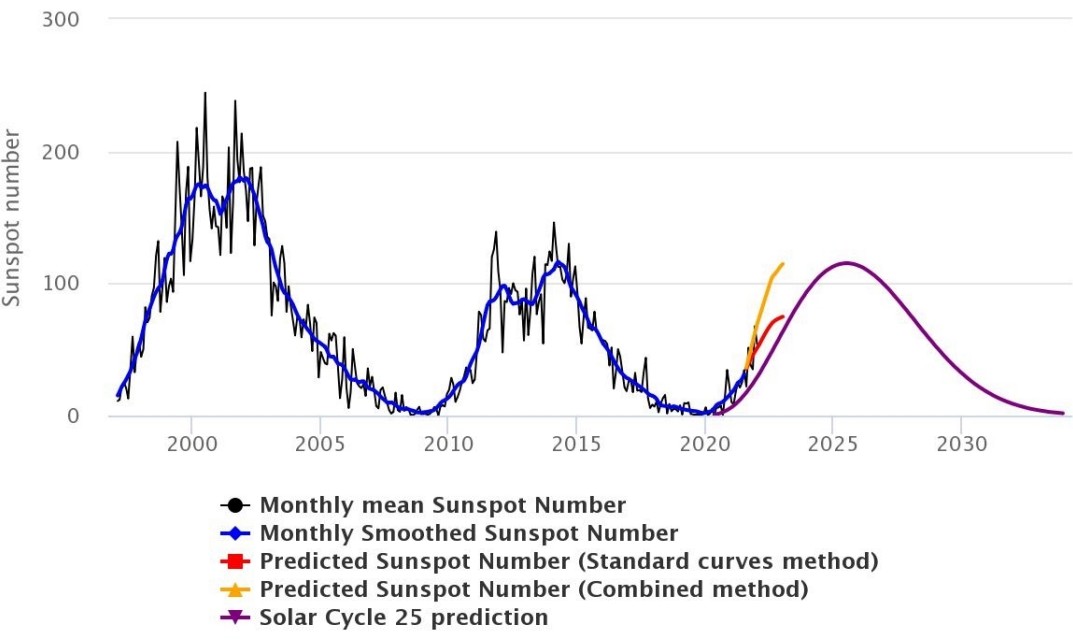

**Figure 11.** Solar cycle progression-Sunspot number.

*7.4. Implementation of Research Results*

1. The Federal State Budgetary Institution of Science, the Institute of Cosmophysical Research and Radio Wave Propagation, Far Eastern Branch of the Russian Academy of Sciences, confirms that in the course of carrying out research work under the RFBR grant No. the program "MMDCSA program—mathematical modeling of the dynamics of solar activity cycles" and registered in Rospatent [39] on 25 December 2019 No. 2019667403.

   The program allows one to simulate the dynamics of solar activity at the stage of rise, to visualize the simulation. The fractional Riccati equation with derivatives of Gerasimov-Caputo fractional orders is taken as a model equation. With the help of a numerical algorithm, solutions of the model are obtained. The 23rd and 24th cycles of solar activity were studied, the model parameters were refined using experimental data, and it was shown that solar activity is decreasing.

2. The Institute of Applied Mathematics and Automation, a branch of the Federal State Budgetary Scientific Institution "Federal Scientific Center" "Kabardino-Balkarian Scientific Center of the Russian Academy of Sciences", confirms that when performing research work on the topic "Mathematical Modeling of the Processes of Geophysics and Physics of Elementary Particles" (No. AAAA -A19-119013190079-5) prepared and published the article [53] D. A. Tverdy "Non-local Cauchy problem for the Riccati equation with a fractional derivative as a mathematical model of solar activity dynamics".

   The paper investigates a mathematical model of solar activity dynamics at the ascent stage, which is based on the Cauchy problem for the Riccati equation of fractional order. The solution obtained by Newton's method is compared with observational data. It is shown that the proposed model is in good agreement with the dynamics of solar activity in the specified period, which allows the identification of trends and detection of memory effects.

## 8. Simulation the Dynamics of Infection with COVID-19

*8.1. Review of Literature and Experimental Data of the Process*

In January 2020, according to WHO, in the Chinese city of Wuhan, the authorities identified a new virus (2019-nCoV) that later became known as SARS-CoV-2. The virus caused pneumonia of unknown etiology.

A month later, on 11 February 2020, ICTV organized an attempt to study the issue from a mathematical point of view, in the work of the authors Tian-Mu Chen, Jia Rui, Qiu-Peng Wang, Ze-Yu Zhao, Jing-An Cui, and Ling Yin [54]. This study aimed to develop a model that mathematically shows the transmissibility of the virus, that is, that the disease is caused by parasites, bacteria or viruses and transmitted by vectors. As a result, it was shown that, probably, bats could act as carriers of infection to humans. The result was also obtained that the transmissibility of SARS-CoV-2 is higher or lower than MERS in various countries and regions of the world.

Since in this study we will carry out graphical modeling of the dynamics of infection of COVID-19, we were interested in the article by Alguliyev, R., Aliguliyev, R. and Yusifov F. [55]. In this scientific work, the authors proposed a conceptual model for the COVID-19 epidemic, which could take into account many factors that could influence the change (growth) in the number of infected people. However, the authors note that this model failed to take into account all factors.

Today, an important task is to study the mathematical foundations of the COVID-19 epidemic, and the subsequent development of various mathematical models of the dynamics of processes associated with COVID-19. Of particular interest are the models based on the mathematical formalism of fractional: differentiation and integration [56–63]. Fractional calculus makes it possible to take into account the heredity of epidemiological processes in the model. The memory effect, as is known, is the property of a dynamic system to remember the previous states of the system. In particular, derivatives of fractional orders allow, for example, to describe that a person with a disease goes through a certain incubation period, and the symptoms may not appear immediately, but after some time the infection will enter the active phase. In other words, the fractional orders will be responsible for the intensity of such a dynamic process.

An analysis of the literature on the topic [56–63] and other sources, leads to the following conclusions:

- the most part these are theoretical works, regarding the existence and uniqueness of a solution to a boundary value problem or the Cauchy problem, for model systems of differential equations, or with different definitions of fractional derivative operators, must be taken into account.;
- the mathematical models under consideration are both difficult to construct a numerical algorithm and further computer implementation;
- as a result, there are no comparisons between model results and experimental data on the COVID-19 epidemic.

In this study [25], we offer a relatively simple mathematical model (5) of the dynamic system of the COVID-19 epidemic, which is based on the fractional Riccati equation, for the following reasons:

- The Riccati equation, as shown in [50], can give good results for describing processes obeying the logistic law, to discover what might be true about the COVID-19 epidemic;
- variable orders of the fractional derivative will allow us to take into account how the effect of variable memory affects the dynamic system. This will allow modeling more complex ongoing epidemiological subprocesses and clarifying the mathematical model.

In this part of the study, which is devoted to the issue of application to the description of the dynamic process of the COVID-19 epidemic, we, using the proposed model (5), will approximate the known data on various processes of the COVID-19 coronavirus epidemic.

In particular, for new cases of COVID-19 infection and for the total number of infected in the Russian Federation, Figures 12 and 13, as well as in the Republic of Uzbekistan, Figures 14 and 15. The experimental data [64] used for comparison was obtained from the "Our World in Data" project with the support of CSSE at JHU.

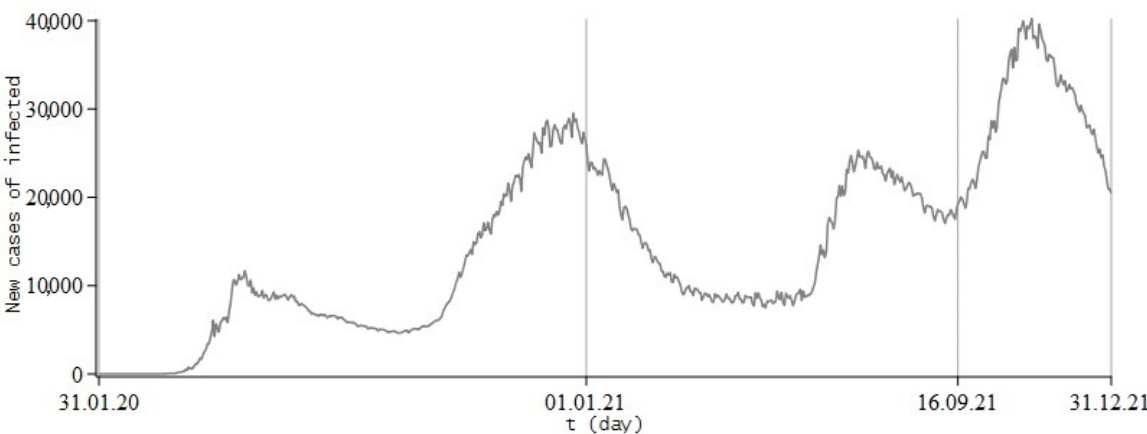

**Figure 12.** Experimental data: for new cases of COVID-19 infection in the Russian Federation, from 31 January 2020 to 31 December 2021, in 1 day increments.

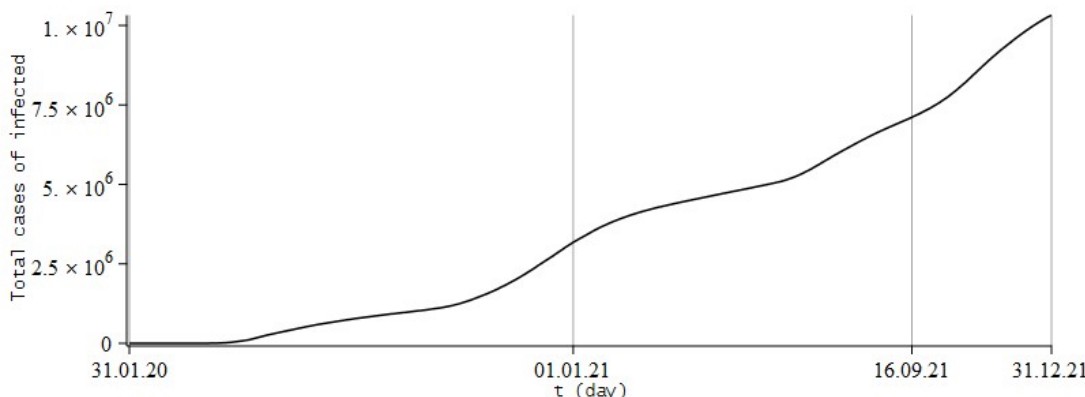

**Figure 13.** Experimental data: for new cases of COVID-19 infection in the Russian Federation, from 31 January 2020 to 31 December 2021, in 1 day increments.

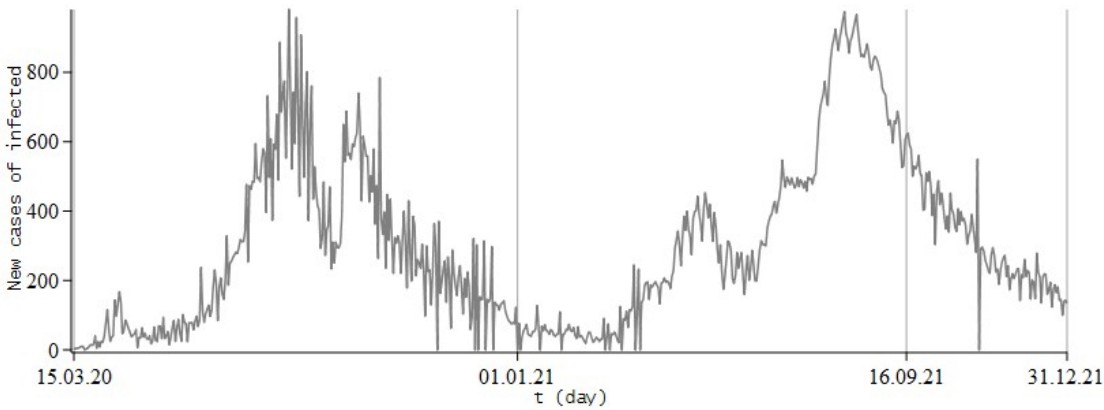

**Figure 14.** Experimental data: for new cases of COVID-19 infection in the Republic of Uzbekistan, from 15 March 2020 to 31 December 2021, with a step of 1 day.

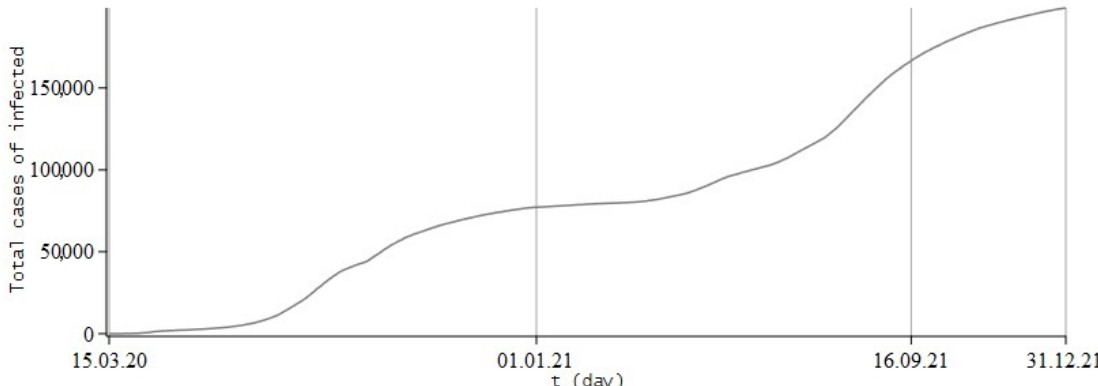

**Figure 15.** Experimental data: for total cases of COVID-19 infection in the Republic of Uzbekistan, from 15 March 2020 to 31 December 2021, with a step of 1 day.

The data is provided as a .xlsx file and is available on the project website ourworldindata. org [64,65]. The file is a table, where each column represents a measurement of one of the observed parameters in increments of one day. The data of this project, with the assistance of JHU, is updated once a day and is publicly available. This publicly available information comes from sources such as national agencies and governments around the world.

Experimental data, as well as the results obtained in the course of numerical simulation, reflect the number of infected people. These data must be coordinated, and therefore nondimensionalized, because mathematics prefers to work only with such quantities, unlike physics. Therefore, normalization is carried out to the maximum, all data on the number of infected people. Further, the results on the vertical axis of the graphs are indicated in relative units.

*8.2. Numerical Experiments for COVID-19*

**Example 2.** *Let us compare the simulation result with data on new cases of infection in the Russian Federation from 31 January 2020 to 16 September 2021, with a step of 1 day (Figure 16, blue curve). We will approximate the input data with the (5) model with parameters:*

$$N = 595, \quad T = 595, \quad u_0 = 0.00006$$

$$\alpha(t) = 0.98 \sin\left(\frac{2.7\pi t}{T}\right)^2, \quad a(t) = 0.8 \sin\left(\frac{2\pi t}{T}\right)^2 - 0.1,$$

$$b = 0.2, \quad c(t) = 0.5 \sin\left(\frac{1.82\pi t}{T}\right)^2 + 0.25.$$

*Let us carry out the proposed model (5), forecasting (Figure 16, red curve) for new cases of infection in the Russian Federation from 16 September to 31 December 2021. Main parameters:*

$$N = 701, \quad T = 701, \quad u_0 = 0.00006,$$

$$\alpha(t) = 0.98 \sin\left(\frac{3.181\pi t}{T}\right)^2, \quad a(t) = 0.8 \sin\left(\frac{2.356\pi t}{T}\right)^2 - 0.1, \tag{11}$$

$$b = 0.2, \quad c(t) = 0.5 \sin\left(\frac{2.14\pi t}{T}\right)^2 + 0.25.$$

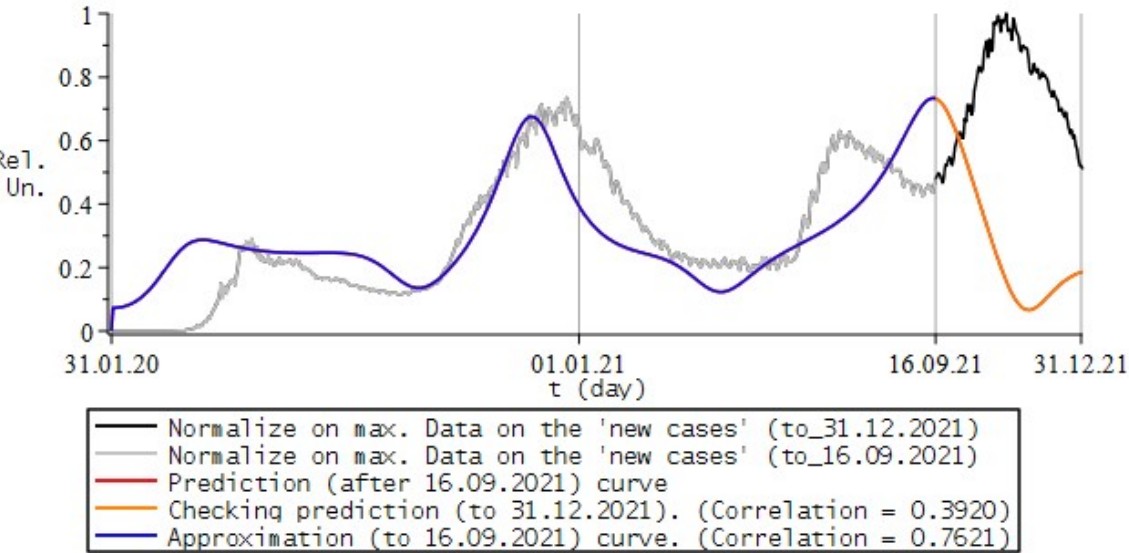

**Figure 16.** Russian Federation. Model curve: (blue)—approximation, with a correlation coefficient: 76.2%; (red)—forecast, through 31 December 2021; (orange)—verification of the forecast with the correlation coefficient: 39.2%. The red curve and the orange curve are obviously the same.

*Now, in this work, let us check how close to reality the predicted (Figure 16, red curve) results turned out to be. To do this, we compare the experimental data until 31 December 2021 and the data obtained by the predictive model (5) with the parameters (11), and also calculate the correlation coefficient between them. The result is visible in (Figure 16, orange curve).*

**Example 3.** *Let us compare the simulation result with the data on the total number of infected in the Russian Federation from 31 January 2020 to 16 September 2021, with a step of 1 day (Figure 17, blue curve). We will approximate the input data with the model (5) with parameters:*

$$N = 595, \quad T = 595, \quad u_0 = 2.812680012 \times 10^{-7},$$

$$\alpha(t) = 0.2, \quad a(t) = 0.2 \sin\left(\frac{3\pi t}{T}\right)^2, \quad b = 0.05, \quad c(t) = 0.25 \sin\left(\frac{0.33\pi t}{T}\right)^2.$$

*Let us carry out the proposed model (5), forecasting (Figure 17, red curve) for new cases of infection in the Russian Federation from 16 September to 31 December 2021. Main parameters:*

$$N = 701, \quad T = 701, \quad u_0 = 2.812680012 \times 10^{-7},$$

$$\alpha(t) = 0.2, \quad a(t) = 0.2 \sin\left(\frac{3.534\pi t}{T}\right)^2, \quad b = 0.05, \quad c(t) = 0.25 \sin\left(\frac{0.3887\pi t}{T}\right)^2. \quad (12)$$

*Now, in this work, let us check how close to reality the predicted (Figure 17, red curve) results turned out to be. To do this, we compare the experimental data until 31 December 2021 and the data obtained by the predictive model (5) with the parameters (12), and also calculate the correlation coefficient between them. The result is visible in (Figure 17, orange curve).*

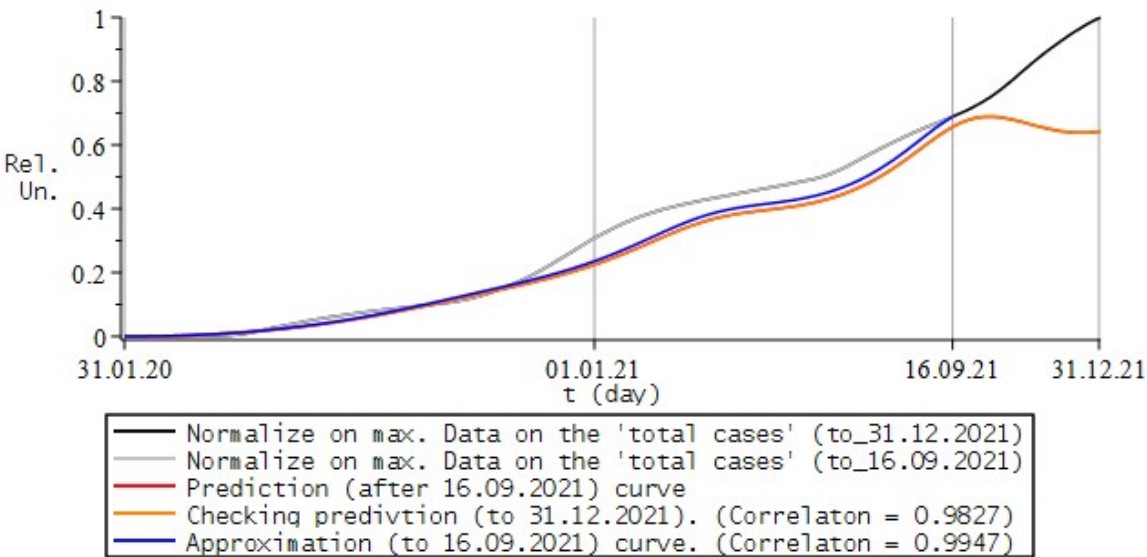

**Figure 17.** Russian Federation. Model curve: (blue)—approximation, with a correlation coefficient: 99.4%; (red)—forecast, through 31 December 2021; (orange)—verification of the forecast with the correlation coefficient: 98.2%. The red curve and the orange curve are obviously the same.

**Example 4.** *Let us compare the simulation result with data on new cases of infection in the Republic of Uzbekistan from 15 March 2020 to 16 September 2021, with a step of 1 day (Figure 18, blue curve). We will approximate the input data with the (5) model with parameters:*

$$N = 551, \quad T = 551, \quad u_0 = 0.0001,$$

$$\alpha(t) = 0.98 \sin\left(\frac{1.64\pi t}{T}\right)^2, \quad a(t) = \frac{t}{T}, \quad b = 0.25, \quad c(t) = 0.5 \sin\left(\frac{\exp(7t/T)t}{T^2}\right)^2$$

*Additionally, we will carry out the proposed model (5), forecasting (Figure 18, red curve) for new cases of infection in the Republic of Uzbekistan from 16 September to 31 December 2021. Main parameters:*

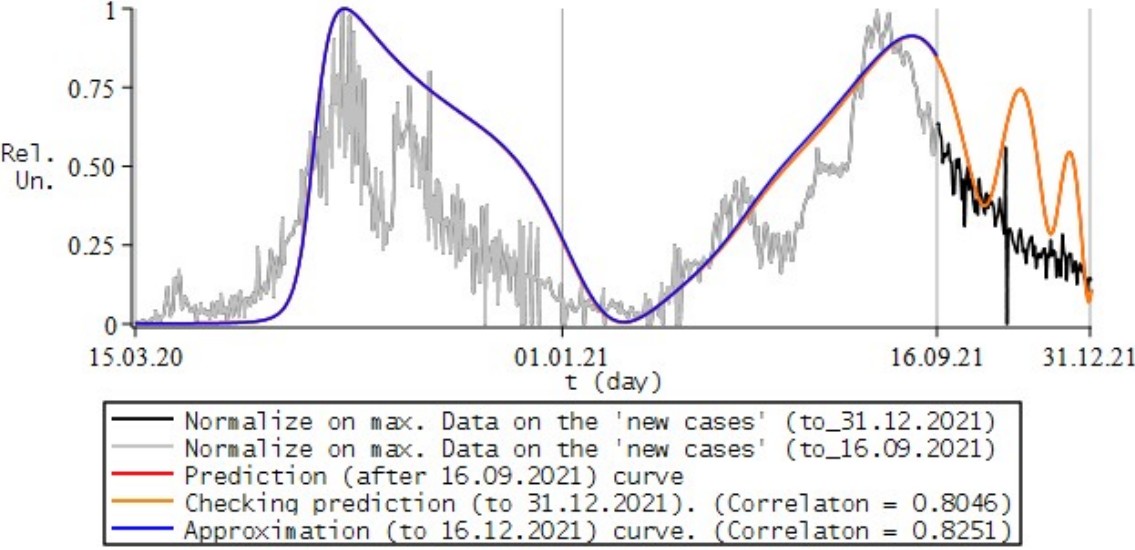

**Figure 18.** The Republic of Uzbekistan. Model curve: (blue)—approximation, with a correlation coefficient: 82.5%; (red)—forecast, through 31 December 2021; (orange)—verification of the forecast with the correlation coefficient: 80.4%. The red curve and the orange curve are obviously the same.

$$N = 657, \quad T = 657, \quad u(0) = 0.0001,$$

$$\alpha(t) = 0.98 \sin\left(\frac{1.96\pi t}{T}\right)^2, \quad a(t) = \frac{1.1923t}{T}, \quad b = 0.25, \quad c(t) = 0.5 \sin\left(\frac{\exp(8.78t/T)t}{T^2}\right)^2 \tag{13}$$

*Now, in this work, let us check how close to reality the predicted (Figure 18, red curve) results turned out to be. To do this, we compare the experimental data until 31 December 2021 and the data obtained by the predictive model (5) with the parameters (13), and also calculate the correlation coefficient between them. The result is visible in (Figure 18, orange curve).*

**Example 5.** *Let us compare the simulation result with the data on the total number of infected people in the Republic of Uzbekistan, from 15 March 2020 to 16 September 2021, with a step of 1 day (Figure 19, blue curve). We will approximate the input by the (5) model, with the parameters:*

$$N = 551, \quad T = 551, \quad u_0 = 6.000816111 \times 10^{-6},$$

$$\alpha(t) = 0.15, \quad a(t) = 0.4 \sin\left(\frac{2\pi t}{T}\right)^2, \quad b = 0.15, \quad c(t) = 0.25 \sin\left(\frac{0.5\pi t}{T}\right)^2.$$

*Let us carry out the proposed model (5), forecasting (Figure 19, red curve) on the total number of infected in the Republic of Uzbekistan from 16 September to 31 December 2021. Main parameters:*

$$N = 657, \quad T = 657, \quad u_0 = 6.000816111 \times 10^{-6},$$

$$\alpha(t) = 0.15, \quad a(t) = 0.4 \sin\left(\frac{2.385\pi t}{T}\right)^2, \quad b = 0.15, \quad c(t) = 0.25 \sin\left(\frac{0.595\pi t}{T}\right)^2. \tag{14}$$

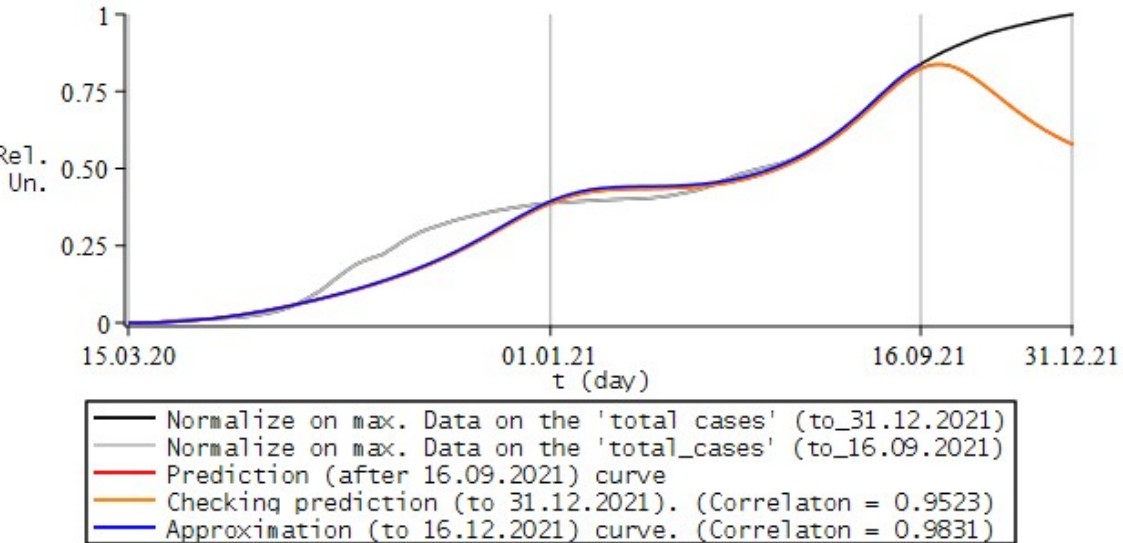

**Figure 19.** The Republic of Uzbekistan. Model curve: (blue)—approximation, with a correlation coefficient: 98.3%; (red)—forecast, through 31 December 2021; (orange)—verification of the forecast with the correlation coefficient: 95.2%. The red curve and the orange curve are obviously the same.

*Now, in this work, let us check how close to reality the predicted (Figure 19, red curve) results turned out to be. To do this, we compare the experimental data until 31 December 2021 and the data obtained by the predictive model (5) with the parameters (14), and also calculate the correlation coefficient between them. The result is visible in (Figure 19, orange curve).*

*8.3. Conclusions and Some Remarks*

The results obtained by the model (5) presented on (Figures 16–19, blue curve), as well as presented on (Figures 16–19, red curve), were previously obtained by the author [25] and published on 3 October 2021.

As can be seen from the Figures 16–19 (blue curve), when choosing the appropriate parameters: $\alpha(t)$ and $a(t), b(t), c(t)$, mathematical model (5), is able to give results close to real data. This indicates the possibility of using fractional equations to describe processes of this type.

This paper shows that our prediction for new infections, from 3 October 2021 (Figures 16 and 18, red curve), visually gives fairly good agreement with experimental data, which is confirmed for the Republic of Uzbekistan in Example 3 on (Figure 18, orange curve) and for the Russian Federation in Example 1 (Figure 16, orange curve).

As a result, we can say that the model, with the selection of appropriate parameters, is able to give some forecast for some parameters of the spread of COVID-19, in particular in terms of the number of new infected people in the countries of the world considered.

Note that the modeling parameters used for (5) may not fully reflect the reality, since:

- the study uses a "phenomenological" approach to show the applicability of fractional calculus, in particular the fractional Riccati equation, to the description of processes of this kind;
- in each of the examples, the model curves were calculated for different variations in the values of the simulation parameters, from which the most appropriate combination was selected according to the maximum correlation coefficient with the experimental ones.

which can be clearly seen in (Figure 16, orange curve), where there is some lag between the model curve and the data being approximated.

In continuation of the work, then it is necessary to compose and solve the corresponding inverse problem in order to clarify the parameters of the mathematical model and their meaning in the context of the problem.

## 9. Simulation the Dynamics of Radon Volumetric Activity in the Accumulation Chamber

*9.1. Formulation of the Problem*

Today, the main idea is that during the preparation of the future earthquake source, changes in the regional stress fields of the matter of the Earth's crust occur, which leads to changes in pressure and temperature gradients, changes (increase) in permeability and porosity. Which, in turn, leads to a change in the rate of radon migration to the surface.

This means that if at the boundary of the lithosphere-atmosphere continuous monitoring of RVA (and other parameters of radon fields) in the subsoil air is conducted with a high degree of detail, then this will make it possible to judge the geophysical processes occurring in the Earth's crust. At least about such processes that could cause a change in the RFD radon flux density from the surface as a source to the atmosphere [66,67], as well as anomalous variations in (recorded) radon concentration. At the same time, one should not forget about the existence of various meteorological and atmospheric values, freezing of the upper layer of soil, and so on, paramaters which affect the flow of radon into the atmosphere.

This suggests the conclusion that measurements of the radon content (more precisely, continuous monitoring of RVA) in the upper soil layer, data processing and subsequent mathematical modeling of RVA are of great interest, for example, in the development of various methods for predicting strong earthquakes. RVA monitoring, for such purposes, has been carried out at the Petropavlovsk-Kamchatka geodynamic test site since 1997 [27].

The concentration of radon is recorded by gas-discharge counters inside storage chambers installed on the ground. For more information on how radon parameters are monitored and the organization of monitoring points, see [27]. Let us just note that the areas of increased radon flow into the atmosphere are characterized by zones with narrow

localization, and their search makes it necessary to quickly evaluate RFD using an accessible method. An example of such a method is also described in detail in [27]. This method was tested in practice in Kamchatka at various points for monitoring subsoil gases [27].

However, when describing the process of radon accumulation in the accumulation chamber, from the point of view of mathematics, in [27], a number of assumptions were made to simplify the solution of the problem, in particular, the accumulation process was assumed to be stationary, that is, when the AER and RFD conditions from the soil surface under the camera, did not change dramatically. This probably means that the factors affecting the speed of this flow were not taken into account, at least atmospheric changes: temperature, pressure and weather conditions. These terms and conditions are expressed in the following model, authored by Firstov, P.P. and Makarov, E.O. [27]:

$$u(t) = u_{max}\left(1 - e^{-\lambda_0 t}\right) + u_0 e^{-\lambda_0 t}. \tag{15}$$

It should be noted that the articles [68,69] provide mathematical modeling of the process of radon transfer in porous soil, due to convection and diffusion, taking into account the hereditary nature of the medium. In some sources, such methods of radon transfer may be called superdiffusion, subdiffusion or anomalous diffusion [13]. We, in this study, will use a slightly different memory effect in the process of radon migration in the chamber.

We, in this work, with the help of mathematical modeling, also investigate the process of radon accumulation in the accumulation chamber in order to determine RFD from the surface. Additionally, in order to more accurately model the accumulation processes, we have improved the existing model, using:

- the introduction of the fractional differential Riccati equation as a model for describing the process of radon accumulation, which will now allow us to take into account heredity, the memory effect of the process;
- introducing a nonlinear function into the model equation, which is responsible for the mechanisms of radon entry into the chamber.

Consider the following law of accumulation of radon volumetric activity RVA (Radon Volumetric Activity) in the chamber:

$$\frac{1}{\Gamma(1-\alpha)} \int_0^t \frac{\dot{u}(\sigma)}{(t-\sigma)^\alpha} d\sigma = F(u, \sigma), \qquad u(0) = u_0, \tag{16}$$

where the left side of the relation is the fractional derivative of the Gerasimov-Caputo constant $0 < \alpha < 1$ [29–31], according to Remark 3, and $\dot{u} = \frac{du}{dt}$.

Now choose a more general function: $F(u, \sigma) = -a(t)u(t)^2 + b(t)u(t) + c(t)$, where $a(t) = c(t)$—functions are continuous on the interval $[0, T]$. Then we arrive at the following Cauchy problem, for the fractional Riccati equation:

$$\partial_{0t}^\alpha u(\sigma) + a(t)u^2(t) - b(t)u(t) - c(t) = 0, \qquad u(0) = u_0, \tag{17}$$

where the term—$a(t)u(t)^2$ of the ratio on the right side, describes the deceleration of radon accumulation in the chamber. More details in the work of the authors [26].

Note that if in (16) we choose the function $F(u, \sigma) = S_D - \lambda_0 u(t)$, where $\lambda_0$ is AER, $S_D = \lambda_0 u_{max}$—RVA mechanism, $u_{max} = \max\limits_t\left(\widehat{u(t)}\right)$—maximum RVA value over time $T$ obtained from experimental data $\widehat{u(t)}$, then we get the results of [70].

Now, let us compare the initial data, the proposed model (17) and the classic model (15). The values on the graphs are given in relative units, since, for the RVA simulation results and input experimental data, normalization to the maximum is carried out.

### 9.2. Numerical Experiment for RVA

To determine the parameters of the model, observational data on radon accumulation at many observation points were used, for example: at the PRTR point (Kamchatka), the USSR point on Sakhalin Island [71] and GL (Kamchatka) [27].

The modeling parameters used were calculated with different variations in the values of the modeling parameters, using the automated procedures described in Section 5.6, from which the most appropriate combination was selected according to the maximum correlation coefficient with the experimental ones.

### 9.3. RVA Modeling Conclusions

As can be seen from the Figures 20–30, the proposed model from [26], which takes into account non-linearity (17), can give results that are closer to real data compared to the known model (15) from [27].

**Example 6** (Viewpoint MP3, camera 1). *We will approximate the input data with general parameters: $N = 46$, $T = 46$, $u_{max} = 1$, $u_0 = 0.2979249012$, $\lambda_0 = 0.1$. The proposed model (17) with $\alpha = 0.95$ with the rest of the parameters: $a = \lambda_0 u_{max}$, $b = 0.001$, $c = \lambda_0 u_{max}$, and the classic model (15).*

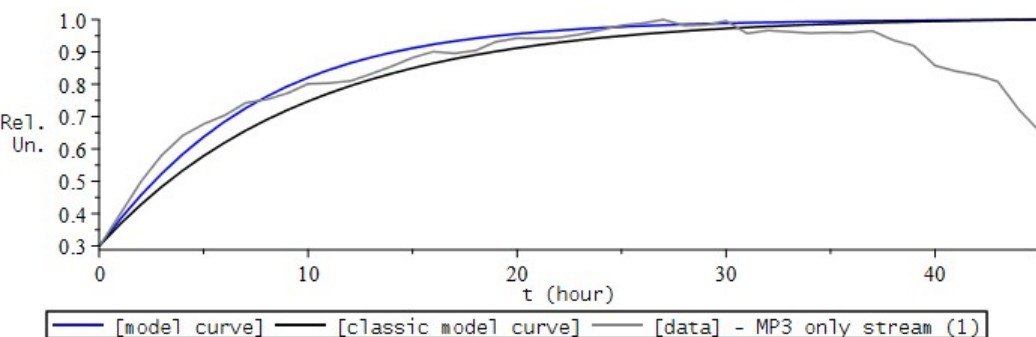

**Figure 20.** Correlation coefficients: 89.9% for the proposed (17) model (blue), and 86.9% for the classic (15) model (black).

**Example 7** (Viewpoint MP3, camera 2). *We will approximate the input data with the parameters: $N = 46$, $T = 46$, $u_{max} = 1$, $u_0 = 0.3353140047$, $\lambda_0 = 0.085$. The proposed model (17) with $\alpha = 0.85$ with other parameters: $a = \lambda_0 u_{max}$, $b = 0.001$, $c = \lambda_0 u_{max}$, and the classic model (15).*

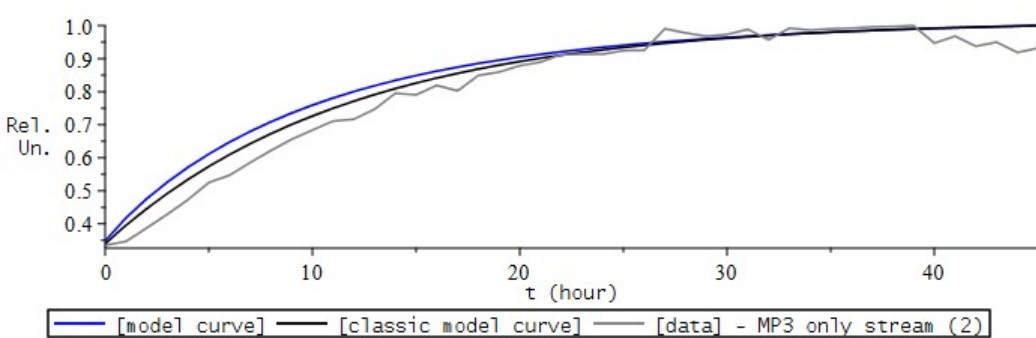

**Figure 21.** Correlation coefficients: 98.6% for the proposed (17) model (blue), and 99% for the classic (15) model (black).

**Example 8.** *Observation point PRT, camera 1. We will approximate the input data with the parameters: $N = 2700$, $T = 45$, $u_{max} = 1$, $u_0 = 0.2554126137$, $\lambda_0 = 0.1$. The proposed model*

*(17) with  alpha = 0.95 with other parameters: $a = \lambda_0 u_{max}$, $b = 0.001$, $c = \lambda_0 u_{max}$, and the classic model (15).*

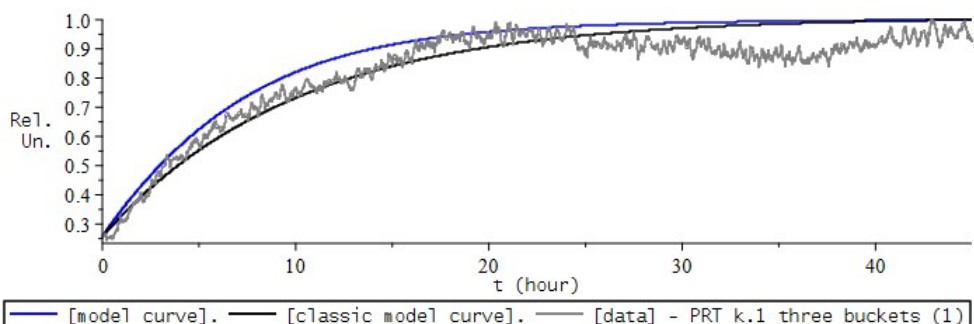

**Figure 22.** Correlation coefficients: 97.9% for the proposed (17) model (blue), and 96% for the classic (15) model (black).

**Example 9** (Observation point PRT, camera 2). *We will approximate the input data with the parameters: $N = 2640$, $T = 44$, $u_{max} = 1$, $u_0 = 0.3813700918$, $\lambda_0 = 0.1$. The proposed model (17) with $\alpha = 0.95$ with other parameters: $a = \lambda_0 u_{max}$, $b = 0.001$, $c = \lambda_0 u_{max}$, and the classic model (15).*

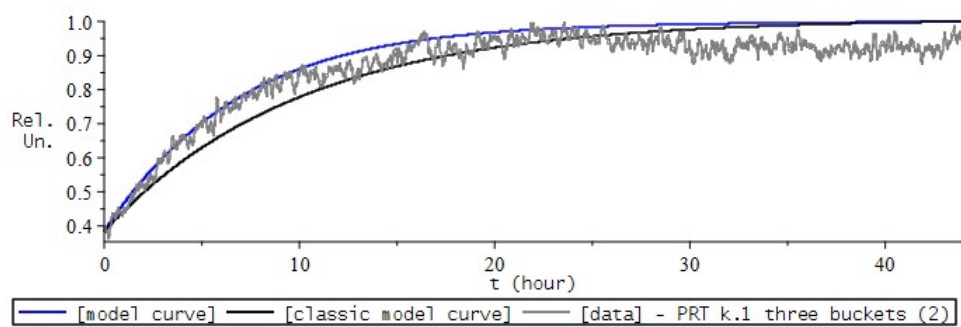

**Figure 23.** Correlation coefficients: 98% for the proposed (17) model (blue), and 95% for the classic (15) model (black).

**Example 10** (Observation point PRT, camera 3). *We will approximate the input data with the parameters: $N = 2700$, $T = 45$, $u_{max} = 1$, $u_0 = 0.3081783500$, $\lambda_0 = 0.1$. The proposed model (17) with $\alpha = 0.95$ with other parameters: $a = \lambda_0 u_{max}$, $b = 0.001$, $c = \lambda_0 u_{max}$, and the classic model (15).*

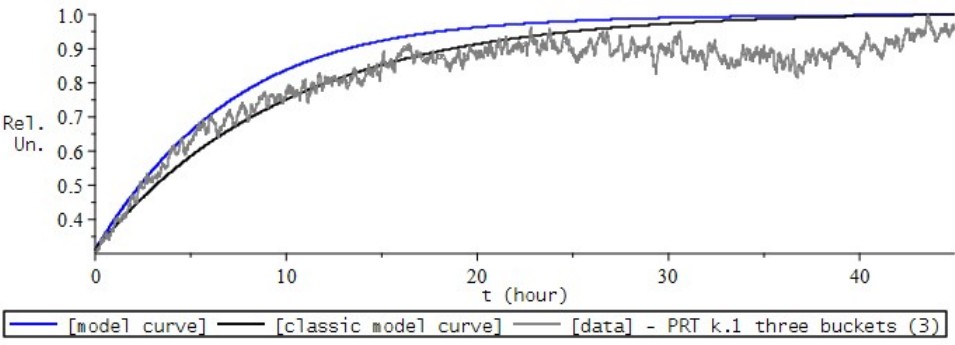

**Figure 24.** Correlation coefficients: 98.2% for the proposed (17) model (blue), and 96.4% for the classic (15) model (black).

**Example 11** (Observation point PRT 21, camera 1). *We will approximate the input data with the parameters:* $N = 97$, $T = 97$, $u_{max} = 1$, $u_0 = 0.7189650259$, $\lambda_0 = 0.045$. *The proposed model (17) with $\alpha = 0.85$ with other parameters:* $a = \lambda_0 u_{max}$, $b = 0.001$, $c = \lambda_0 u_{max}$, *and the classic model (15).*

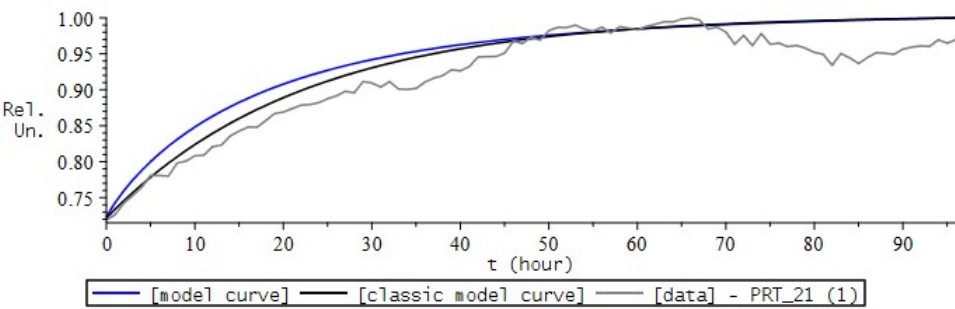

**Figure 25.** Correlation coefficients: 96.1% for the proposed (17) model (blue), and 96.8% for the classic (15) model (black).

**Example 12** (Observation point PRT 21 only stream, camera 1). *We will approximate the input data with the parameters:* $N = 121$, $T = 121$, $u_{max} = 1$, $u_0 = 0.7189650259$, $\lambda_0 = 0.045$. *The proposed model (17) with $\alpha = 0.85$ with other parameters:* $a = \lambda_0 u_{max}$, $b = 0.001$, $c = \lambda_0 u_{max}$, *and the classic model (15).*

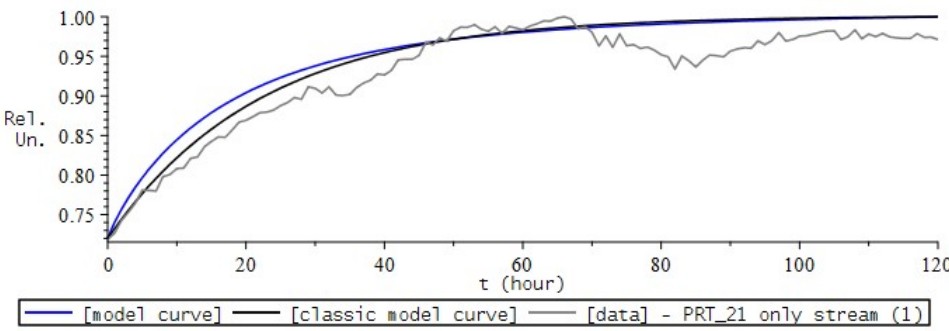

**Figure 26.** Correlation coefficients: 96.5% for the proposed (17) model (blue), and 97.1% for the classic (15) model (black).

**Example 13** (Observation point PRT 21 t stream, camera 2). *We will approximate the input data with the parameters:* $N = 121$, $T = 121$, $u_{max} = 1$, $u_0 = 0.8117815770$, $\lambda_0 = 0.04$. *The proposed model (17) with $\alpha = 0.8$ with other parameters:* $a = \lambda_0 u_{max}$, $b = 0.001$, $c = \lambda_0 u_{max}$, *and the classic model (15).*

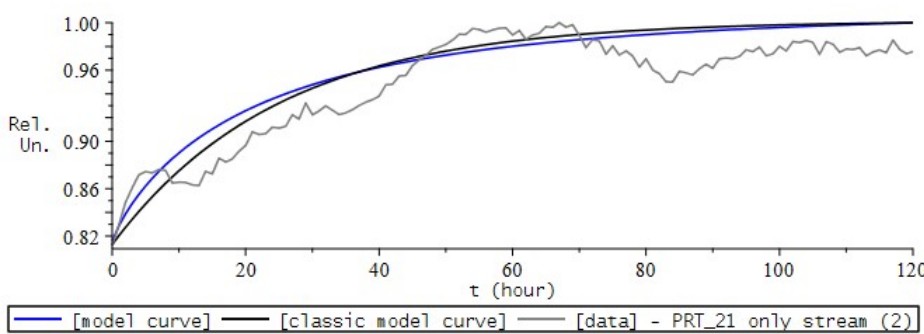

**Figure 27.** Correlation coefficients: 92.5% for the proposed (17) model (blue), and 93.7% for the classic (15) model (black).

**Example 14** (Sakhalin observation point, camera 1)**.** *We will approximate the input data with the parameters:* $N = 320$, $T = 160$, $u_{max} = 1$, $u_0 = 0.2383330133$, $\lambda_0 = 0.06$. *The proposed model* (17) *with* $\alpha = 0.9$ *with other parameters:* $a = \lambda_0 u_{max}$, $b = 0.001$, $c = \lambda_0 u_{max}$, *and the classic model* (15).

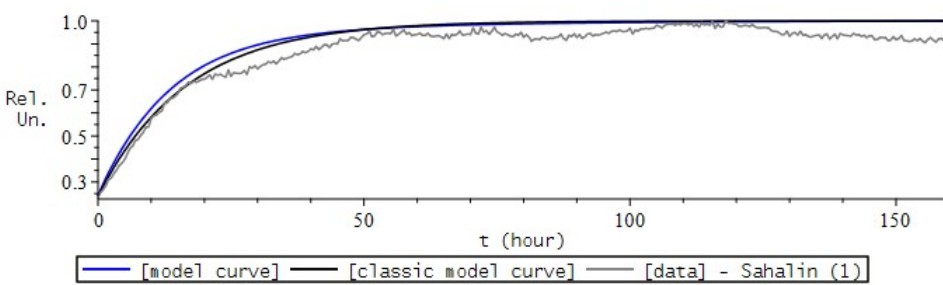

**Figure 28.** Correlation coefficients: 98.6% for the proposed (17) model (blue), and 98.9% for the classic (15) model (black).

**Example 15** (Observation point GLL stream, camera 2)**.** *We will approximate the input data with the parameters:* $N = 657$, $T = 1314$, $u_{max} = 1$, $u_0 = 0.2407407407$, $\lambda_0 = 0.01$. *The proposed model* (17) *with* $\alpha = 0.85$ *with other parameters:* $a = \lambda_0 u_{max}$, $b = 0.001$, $c = \lambda_0 u_{max}$, *and the classic model* (15).

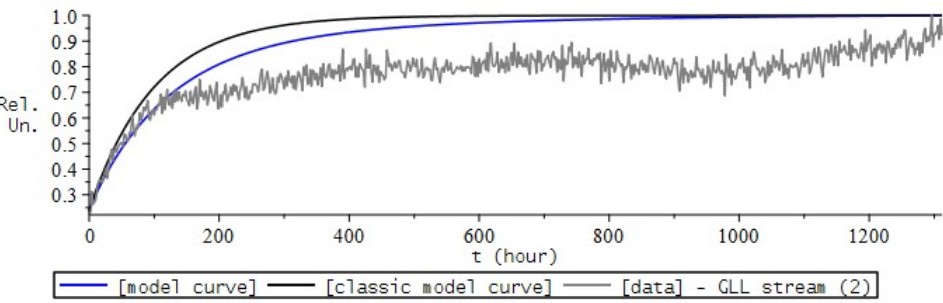

**Figure 29.** Correlation coefficients: 92% for the proposed (17) model (blue), and 90% for the classic (15) model (black).

**Example 16** (Observation point GLL stream, camera 1)**.** *We will approximate the input data with the parameters:* $N = 657$, $T = 1314$, $u_{max} = 1$, $u_0 = 0.4844827586$, $\lambda_0 = 0.1$. *The proposed model* (17) *with* $\alpha = 0.25$ *with other parameters:* $a(t) = \frac{(1 - \lambda_0 u_{max})t}{T}$, $b = 0.001$, $c(t) = \frac{(1 - \lambda_0 u_{max})t}{T}$, *and the classic model* (15).

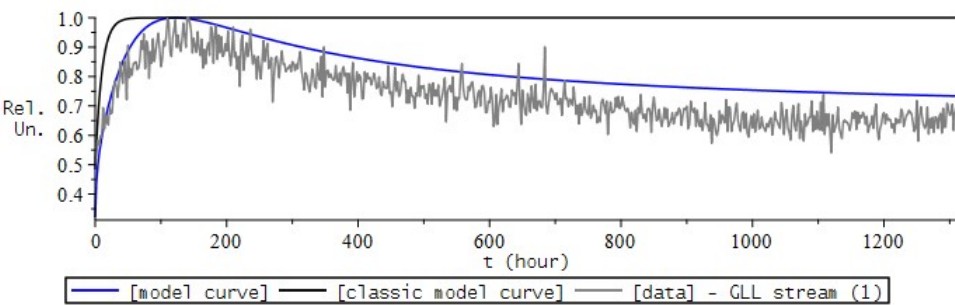

**Figure 30.** Correlation coefficient: 89.7% for the proposed (17) model (blue), and 17% for the classic (15) model (black).

## 10. Conclusions

Various possible applications of the fractional Riccati equation for modeling dynamic processes with saturation and memory effect have been proposed. In particular: SA speakers, RVA speakers in a storage room, as well as the dynamics of infection with COVID-19 in different countries around the world. With the help of the developed library in the Maple 2021 symbolic mathematics environment, the above processes were simulated. The parameters of the proposed mathematical model were selected in the best way based on comparison with smoothed and complete experimental data of these processes. Good results were obtained by comparing the obtained model curves and complete experimental data. This indicates that, as a model equation, the fractional Riccati equation with a VO derivative of the Gerasimov-Caputo type and non-constant coefficients as functions is adequate and applicable to saturation processes taking into account dynamic memory.

Theoretical results [24] on numerical methods of solution, the results of mathematical modeling of some physical processes, as well as the developed programs, were obtained, carried out and created, within the framework of the PhD dissertation in physical and mathematical sciences. All three of these points are necessary requirements for the scientific novelty of the dissertation, for successful defense and obtaining a PHd degree. The introduction mentions that there are a number of gaps in the literature on this topic, and this PhD thesis is devoted to filling these gaps, and the results constitute a scientific novelty.

## 11. Patents

As part of the study "Mathematical modeling using the fractional Riccati equation of some dynamic processes with saturation and memory effects", which includes the work [24] and this article, for the Maple 2021 symbolic computer mathematics environment, the «FDRExt» library was developed. This program code implements: numerical methods and algorithms required for model calculations (5); algorithms for numerical analysis of the numerical methods used; algorithms for extracting, processing and recording experimental and calculated data for their comparison; functions for data visualization.

**Author Contributions:** Conceptualization, R.P.; methodology, R.P.; software, D.T.; validation, D.T.; formal analysis, D.T.; investigation, D.T.; resources, D.T. and R.P.; data curation, D.T.; writing—original draft preparation, D.T.; writing—review and editing, D.T.; visualization, D.T.; supervision, R.P.; project administration, R.P.; funding acquisition, R.P. All authors have read and agreed to the published version of the manuscript.

**Funding:** The work was carried out within the framework of the state assignment on the topic "Physical processes in the system of near space and geospheres under solar and lithospheric influences" (No. AAAA-A21-121011290003-0). The work was carried out within the framework of a grant from the President of the Russian Federation Development of mathematical models of fractional dynamics for the purpose of studying oscillatory processes and processes with saturation MD-758.2022.1.1.

**Institutional Review Board Statement:** Not applicable.

**Informed Consent Statement:** Not applicable.

**Data Availability Statement:** Not applicable.

**Acknowledgments:** The authors are grateful to Mamchuev Murat Osmanovich for valuable comments and discussion of the results.

**Conflicts of Interest:** The author declares no conflict of interest.

## Abbreviations

The following abbreviations are used in this manuscript:

| | |
|---|---|
| VO | variable order |
| EFDS | Explicit Finite-difference Method |

|       |                                                    |
|-------|----------------------------------------------------|
| IFDS  | Implicit Finite-difference Method                  |
| ONM   | ordinary Newton Method                             |
| MNM   | Modified Newton Method                             |
| UV    | Ultraviolet                                        |
| SA    | Solar Activity                                     |
| WHO   | World Health Organization                          |
| SARS-CoV-2 | Severe acute respiratory syndrome-related coronavirus 2 |
| ICTV  | International Committee on Taxonomy of Viruses      |
| MERS  | Middle East Respiratory Syndrome                   |
| COVID-19 | COronaVIrus Disease 2019                        |
| CSSE  | Center for Systems Science and Engineering         |
| JHU   | Johns Hopkins University                           |
| RVA   | Radon Volumetric Activity ($Bq/m^3$)               |
| Rn    | Radon                                              |
| RFD   | Radon Flux Density                                 |
| AER   | Air Exchange Rate                                  |

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
