# Peer review of "Application of the Fractional Riccati Equation for Mathematical Modeling of Dynamic Processes with Saturation and Memory Effect"

_fractalfract, doi:10.3390/fractalfract6030163_

Round 1

Reviewer 1 Report

The report is attached.

Author Response

Answers to the review in the file.

Reviewer 2 Report

The authors presented the results of an extensive experimental study of the application of the fractional Riccati equation for modeling various dynamic processes.
The proposed method and the software described in detail are of scientific value and can be widely used in practice.
A few suggestions for improving the manuscript.
1. The article contains many sections. Moreover, sections 7,8 and 9 contain the results of experiments with different physical systems and processes. I propose to clearly describe the structures of the article at the end of section 1.
2. Section 5 contains a detailed description of the developed software, including listings. However, not all readers may be fluent in the MAPLE environment. I propose to present a generalized flowchart of the developed software. This is necessary both for a clearer understanding of the essence of the proposed algorithms, and to increase reader interest in the article.

Author Response

Answers to the review in the file.

Reviewer 3 Report

The paper is very interesting and of good quality. I have some suggestions for improvement. 

1/ please indicate again clearly your scientific contribution and motivation with respect to existing literature. 

2/ explain the main aim of variable order derivative in front of fractional derivative.

3/ Figure 1 is not appearing, please check.

4/ the whole content of the work is very big for a research paper:

26 Figures and 87 References, please minimize and reduce to have a nice readable piece. 

5/ the codes/sources of numerical/simulation techniques not necessary to be appeared,  it is needed the obtained results. 

6/ you have the option to rename this work as Survey. 

Author Response

Answers to the review in the file.

Reviewer 4 Report

The manuscript presents the application of a model Riccati equation based on variable-order (VO) fractional calculus to different real-life physical phenomena. In general, the paper is well-written and rigorous. This reviewer has the following suggestions and concerns:

1) How was the inverse problem solved? More specifically, in applying the VO Riccati model to the different physical phenomena, how was the VO \alpha(t) determined? This aspect is not clear from the manuscript.

2) In regards to 1) additionally, how were the variable coefficients in the Riccati model determined for the different applications?

3) The dimensional consistency of the VO Riccati model in Eq. (5) is not clear. This indicates a serious flaw in the manuscript and should be corrected. Recall that introduction of fractional derivatives leads to fractional-order units which must be corrected using appropriate dimensional constants that also have a solid physical significance.

4) The introduction can be enriched with the following papers which describe the state of the art of variable-order fractional calculus: (a) doi.org/10.1098/rspa.2019.0498; and (b) doi.org/10.1016/j.physa.2009.07.024

5) Although the manuscript is well-written, this reviewer could find several grammar and typesetting errors (see the line 334 , for example). Please proofread again.

Author Response

Answers to the review in the file.

Round 2

Reviewer 4 Report

The manuscript can be accepted for publication in the present form.

This manuscript is a resubmission of an earlier submission. The following is a list of the peer review reports and author responses from that submission.